# LipFed: Mitigating Subgroup Bias in Federated Learning with Lipschitz Constraints

## Abstract

Federated learning (FL) is a promising paradigm for training decentralized machine learning models with privacy preservation. However, FL models are biased, leading to unfair model outcomes towards subgroups with intersecting attributes. To address this, we propose LipFed, a subgroup bias mitigation technique that leverages Lipschitz-based fairness constraints to mitigate subgroup bias in FL. We evaluate LipFed's efficacy in achieving subgroup fairness across clients while preserving model utility. Our experiments on benchmark datasets and real-world datasets demonstrate that LipFed effectively mitigates subgroup bias without significantly compromising group fairness or model performance.

## 1 Introduction

Federated learning (FL) trains a global model using decentralized edge devices' private data without collecting their data centrally, promoting collaborative learning while preserving data privacy McMahan et al. (2017). This makes FL suitable for privacy-sensitive applications such as medical diagnosis Feki et al. (2021); Ku et al. (2022), gender prediction Krishnan et al. (2020), next-character prediction Sun et al. (2022), and activity recognition Ek et al. (2020); Ouyang et al. (2021); Sozinov et al. (2018). Despite collaborative learning and privacy preservation benefits, FL inevitably learns undesired biases from statistically heterogeneous clients' data Abay et al. (2020). For instance, a crime detection FL algorithm may predict crime suspects based on skin color Courtland (2018), leading to the wrongful prediction of who goes to jail Polonski (2018). Unchecked biases in FL can erode user trust and negatively impact user experiences, affecting FL adoption and acceptance.

Recent FL research has focused on addressing bias, targeting *individual bias* Li et al. (2019a); Mohri et al. (2019); Deng et al. (2020); Li et al. (2020); Hu et al. (2022); Horvath et al. (2021) and *group bias* Yue et al. (2021); Cui et al. (2021); Papadaki et al. (2022). Individual bias techniques aim to ensure similar model performance across clients Papadaki et al. (2022), with approaches like Mohri et al. (2019); Deng et al. (2020); McMahan et al. (2017) optimizing the worst-performing client's performance through importance weighting. In contrast, group fairness techniques (*fairness across multiple sensitive attributes* Wang et al. (2020)) enforce fairness constraints for individual attributes like race, gender, or label Papadaki et al. (2022); Chen et al. (2022). However, they often do not guarantee fairness for subgroups with intersecting characteristics, performing well for some while failing others, as discussed in **Example 1** and shown in Figure 1.

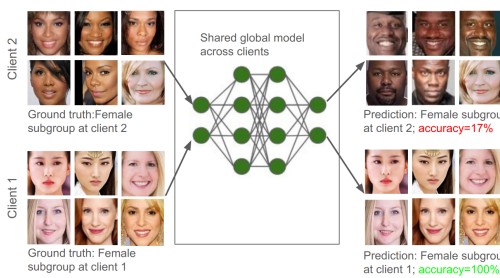

Figure 1: Subgroup Bias in FL. The global model achieves 100% accuracy on Client 1's diverse subgroup but only 17% on Client 2's predominantly black women subgroup, highlighting bias from uneven data distribution across clients

**Example 1.** *In a hypothetical scenario with race (black and white) and gender (male and female) as groups, consider a classifier predicting positive outcomes only for black men or white women. This classifier appears fair across groups, predicting positively for both men and women $50\%$ and both black and white groups $50\%$ of the time. However, examining subgroups like black and white women violates statistical parity fairness. For instance, black women may be disproportionately labeled unfavorably, causing an unfair disadvantage for this intersectional subgroup. This example demonstrates Simpson's Paradox Pearl (2022) in fairness evaluation, where seemingly fair techniques for groups become unfair for their fine-grained subgroups.*

In FL, *subgroup bias* arises because subgroups across clients fail to be *independent and identically distributed (IID)*. This deviation from IID-ness happens because subgroups across clients can have diverse *feature distributions* due to factors such as geographical location, weather, and data collection devices Hsieh et al. (2020); Lyu et al. (2020). For example, images of *black and white female* faces can vary dramatically worldwide due to their skin color. Images of black and white female faces can also look very different given the quality of their collection devices; low vs. high-quality collection devices, etc. *Subgroup fairness* is vital because it reveals hidden biases within intersecting groups, as illustrated in the previous example. Ensuring fairness across both intersectional subgroups and broad groups is necessary to avoid biases. Based on this requirement, our research question is: ***how can FL models achieve subgroup fairness without compromising overall group fairness and model utility?***

To address subgroup fairness in FL, we propose *Lipschitz Fair Federated Learning (LipFed)*, a novel framework applying the *Lipschitz property* Dwork et al. (2012) to decentralized FL. While Lipschitz constraints have been used before (§B.1), our approach uniquely adapts them to ensure equitable model performance across diverse subgroups on different devices. LipFed leverages a distance metric to measure subgroup similarity and performance distributions across clients, overcoming the complexity of decentralized data(§4.1).

**Contributions.** In summary, we make the following contributions:

1. We identify the subgroup bias problem in FL (§3), focusing on bias at the subgroup level rather than statistical bias across fixed demographic groups, addressing intersectional biases more comprehensively, ensuring fairness, and reducing discrimination based on intersecting attributes.
2. We propose LipFed, leveraging the Lipschitz property to train subgroup fair models in FL, ensuring minor changes in sensitive features lead to minor changes in model predictions, thus promoting fair subgroup outcomes in FL models.
3. We conduct theoretical analysis and establish precise bounds for subgroup and statistical fairness. By providing clear bounds (§4.2, §C.1), our work promotes a more transparent and accountable approach to addressing subgroup and statistical fairness challenges, fostering trust and reliability in FL.
4. We apply the LipFed across datasets (§5), reducing subgroup bias by up to 49% without degrading model utility, though with some trade-offs in statistical fairness, clarified through our theoretical analysis (§C.2). LipFed also improves other existing FL methods, by up to 25% in mitigating subgroup bias.

## 2 RELATED WORK

This section reviews methods in FL fairness, focusing on those related to subgroup fairness. While subgroup fairness is recognized in centralized learning, we address the unique challenges of FL and highlight the limitations of existing approaches. Due to space constraints, additional methods are discussed in Appendix D.

FL algorithms aimed at achieving a globally fair model are typically classified into three distinct categories, including *client-fairness*Li et al. (2019a); Mohri et al. (2019); Deng et al. (2020); Li et al. (2020); Hu et al. (2022); Horvath et al. (2021), *group-fairness*Yue et al. (2021); Cui et al. (2021); Papadaki et al. (2022); Selialia et al. (2023), and *robustness techniques* Lee et al. (2022); Karimireddy et al. (2020).

**Client fairness.** Ensuring fairness among clients in FL is essential to mitigate biases from non-IID data distributions across devices. Techniques such as Federated Fair Averaging (FedFV) Wang et al. (2021) adjust gradient directions and magnitudes to balance model *average performance* based on client contribu-

tions Papadaki et al. (2022), while GIFair-FL Yue et al. (2023) dynamically modifies model updates with a fairness-aware aggregator to reduce *average loss*. FjORD Horvath et al. (2021) employs ordered dropout to customize model sizes to client capacities, enhancing both fairness and accuracy. Additionally, Agnostic Federated Learning (AFL) Mohri et al. (2019) tailors the global model to any client distribution mix, q-FFL Li et al. (2019a) reweights losses to favor lower-performing devices, and Tilted Empirical Risk Minimization (TERM) Li et al. (2020) fine-tunes outlier impact and class representation, collectively improving *average performance* in diverse environments.

**Group fairness.** Recent advancements in FL emphasize addressing group fairness and biases against protected groups. FairFed Ezzeldin et al. (2023) uses fairness-aware aggregation and local debiasing to enhance group fairness under heterogeneous data conditions. FedMinMax Papadaki et al. (2022) employs alternating optimization for minimax fairness across demographic groups, showing competitive performance. FCFL Cui et al. (2021) combines algorithmic fairness and performance consistency, achieving Pareto optimality via gradient-based methods and outperforming existing models in fairness and utility.

**Limitations of existing techniques.** While valuable, current bias mitigation techniques in FL do not ensure fairness for subgroups with overlapping characteristics. According to Simpson's Paradox Pearl (2022), seemingly group-fair techniques may still exhibit unfair outcomes towards fine-grained subgroups. The following section explores these deviations and their implications through empirical studies.

## 3 PRELIMINARIES AND PROBLEM FORMULATION

This section defines formal definitions of FL and the problem of *subgroup fairness* addressed in this paper, establishing the study's framework. Specifically, this section covers the local data heterogeneity of decentralized FL clients, how FL learns from such heterogeneous data across clients, and subgroup fairness in FL. In this section, the key question we aim to answer through an empirical study is: *what is the effect of data heterogeneity on subgroup fairness across clients in FL?*

### 3.1 PRELIMINARIES

**Federated learning (FL).** trains a global model using a server and $K$ decentralized clients, ensuring privacy by not sharing their local data. Each client $k \in K$ has its private local dataset $\mathscr{D}_k = \{\boldsymbol{X}_k, \boldsymbol{Y}_k\}$, with $N_k$ tuples $\{(\boldsymbol{x}_k^n \in \boldsymbol{X}_k, y_k \in \boldsymbol{Y}_k^n)\}_{n=1}^{N_k}$ representing input and output spaces. These private datasets can be grouped by attributes like race, gender, or label Chen et al. (2022). The local group dataset on client $k$ is $\mathscr{D}_{g,k} = \{\boldsymbol{X}_k^g, \boldsymbol{Y}_k^g\}^{N_g}$ with $N_g \leq N_k$ samples where $g \in G$ indicates group membership. In ideal IID scenarios, clients sample $\mathscr{D}_{g,k}$ independently from a global distribution $f_g(\boldsymbol{X})$. However, real-world FL scenarios often feature non-IID/heterogeneous data due to factors like *inter-partition decorrelation* Hsieh et al. (2020); Liu et al. (2020), which occurs when clients fail to share standard specifications/features, resulting in decorrelated local group data across clients.

**Subgroups.** FL aggregates non-IID local group data from decentralized clients into a unified dataset, $\mathscr{D} = \bigcup_{k=1}^K \mathscr{D}_k$, representing global groups from multiple sources. Each client's local data $\mathscr{D}_k$ includes unique local groups $\mathscr{D}_{g,k}$. Thus, $\mathscr{D}$ integrates these groups, and each global group (e.g., females) includes local group structures from all clients. We refer to these local groups as *subgroups* of that global group within the unified data representation. FL uses the unified dataset $\mathscr{D}$ to learn an optimal global model $h^*$ (with global parameters $\boldsymbol{\theta}$) from a class of hypotheses $H$ that map input features $\boldsymbol{x}_k^n$ to outputs $y_k^n$. The optimal model minimizes the *empirical risk* objective with $F_k$ as the empirical risk for client $k$ with local parameters $\boldsymbol{\theta}_k$ as:

$$\boldsymbol{\theta}^* = \arg\min_{\boldsymbol{\theta}} \left\{ R(\cdot; \boldsymbol{\theta}) = \sum_{k=1}^K \left( \frac{N_k}{\sum_{k=1}^K N_k} \right) R_k(h_{\boldsymbol{\theta}_k}(\boldsymbol{X}_k), \boldsymbol{Y}_k) \right\} \tag{1}$$

**Subgroup fairness in FL.** Many FL works aim to achieve a modified formulation of Equation 1 for group-fair model parameters Mohri et al. (2019); Yue et al. (2023); Li et al. (2020), often overlooking subgroup fairness.

Suppose there are $n_k$ subgroups $\{g_k\}_{k=1}^{N_k}$ within a group $g$. Let the performance measures of models $h_1$ and model $h_2$ for these subgroups be represented as true positive rates (TPR), be $\{a_1^{g_{g,k}}\}_{k=1}^{N_k}$ and $\{a_2^{g_{g,k}}\}_{k=1}^{N_k}$, respectively. Model $h_1$ is more subgroup fair than model $h_2$ if $Disc_{h_1}(\{a_1^{g_{g,k}}\}_{k=1}^{n_k}) < Disc_{h_2}(\{a_2^{g_{g,k}}\}_{k=1}^{N_k})$, where performance discrepancy $Disc_h$ is calculated as (detailed theory in §F.2):

$$Disc_h(\{a^{g,k}\}_{k=1}^{N_k}) = \max\{a^{g,k} - a^{g,k'}\} \quad \forall k, k' \in K; k \neq k' \tag{2}$$

Higher performance discrepancy indicates greater variation in subgroup performance metrics, indicating potential bias. Performance is measured using $TPR_g = \frac{TP_g}{TP_g + FN_g}$ from fairness-aware optimization in FL Poulain et al. (2023) where $TP_g$ counts true positives (correctly classified instances) and $FN_g$ counts false negatives (incorrectly classified instances) for group $g$ (theory in §F.1).

**Lipschitz fairness.** Achieving individual-level fairness across similar entities $x$ and $x'$, where the similarity of these entities is quantified by the distance metric $d(x, x')$, can be done by optimizing the model to satisfy the Lipschitz property Dwork et al. (2012).

*Definition 3.1 (Lipschitz model).* A model $h_{\theta} : G \rightarrow \Delta(A)$ satisfies the $(D, d)$-Lipschitz property if for every $x, x' \in G \quad \exists \epsilon > 0$ such that:

$$D(h_{\theta}(x), h_{\theta}(x')) \leq \epsilon \cdot d(x, x'). \tag{3}$$

Here, $d : G \times G \longrightarrow \mathbb{R}$ quantifies the similarity between individuals. Without a well-defined metric, $d(\cdot)$ reflects the "best" available approximation agreed upon by society Dwork et al. (2012). $h_{\theta} : G \rightarrow \Delta(A)$ maps individual samples to outcomes (e.g., an individual's TPR).

**Intuition.** In diverse FL edge deployments with non-IID local data, the global model aggregated via FedAvg McMahan et al. (2017) can converge to an unfair model towards subgroups in a group across clients. But the Lipschitz condition in Equation 3 requires that similar individuals $x, x'$ should have outputs $h_{\theta}(x)$ and $h_{\theta}(x')$ with the Euclidean distance $D(h_{\theta}(x), h_{\theta}(x'))$ between $h_{\theta}(x)$ and $h_{\theta}(x')$ is at most $d(x, x')$.

### 3.2 EXPERIMENTAL SETUP

To examine the impact of non-IID data on subgroup bias in FL, we conduct experiments on image classification using FedAvg to aggregate local models. We use four deep learning models across six datasets (two benchmarks, two real-world, and two fairness-based and large-scale), partitioned based on non-IID features across $K = 5, 10$ clients. For model setup, ResNet He et al. (2016) is applied to FER2013 Giannopoulos et al. (2018) for emotion recognition (grouping seven emotions Papadaki et al. (2022)), LeNet LeCun et al. (1998) for MNIST Baldominos et al. (2019) (with each digit as a group), VGGNet Dhillon & Verma (2020) for FashionMNIST Xiao et al. (2017) (with each product as a group), ResNet for UTK Savchenko (2021) (for gender prediction), and Logistic Regression Hosmer et al. (1997) for two ACS datasets Ding et al. (2021) (for income: ASCI and employment prediction: ASCE). For ASCI, data is distributed by state to form two groups (Income True/False), with the state acting as an implicit sensitive attribute. For ASCE, data is filtered for individuals aged 16 to 90, forming employed/unemployed groups (see §E.3).

*Note: Though our experiments involve a limited number of datasets and clients, the theoretical guarantees in C ensure that LipFed's fairness and utility scale are reliable for the scope of the academic paper. These guarantees validate the robustness of our approach, even in broader FL settings.*

**Data Partitions.** Benchmark and real-world datasets are partitioned across clients using a Dirichlet distribution Hsu et al. (2019); Wang et al. (2020). For the income and employment tasks, data is naturally partitioned across approximately 50 clients, allowing us to validate the scalability of our approach in more complex settings (more details about experimental setup can be found in E.

**Heterogeneous feature distributions.** We simulate feature noise in image data with Gaussian distribution ($\tilde{I}(x, y) = I(x, y) + \epsilon$, where $\epsilon \sim \mathcal{N}(0, \sigma^2)$) to explore bias in global models due to non-IID subgroup data diverging from pristine distributions, controlling noise intensity through variance $\sigma^2$, with $\sigma \geq 0.03$

mimicking real-world conditions Ghosh et al. (2018); Saenko et al. (2010); Song et al. (2022); Lyu et al. (2020). Concurrently, the ACS fairness dataset, partitioned by state, captures unique demographic landscapes reflecting inherent feature heterogeneity in socio-economic factors like age, education, race, and occupation, where average income, education levels, and employment rates vary significantly across states.

**Evaluation metrics.** Bias mitigation aims to minimize discrepancies (for subgroups) while maintaining competitive utility; in doing so, we assess three key metrics (additional discussion in §F):

*Subgroup bias metrics.* measure performance discrepancies across subgroups. We compute subgroup discrepancies $\{Disc_h(\{a^{g,k}\}_{g=1}^G)\}$ for each global group $g$, where $a^{g,k}$ is the model's performance on each subgroup. We compare the distribution of these subgroup discrepancies across global groups using their median $M$ values. Low median values (approaching zero) indicate low subgroup bias.

*Group bias metrics.* measure performance discrepancies across groups. We compute the discrepancies $\{Disc_h(\{a^g\}_{g=1}^G)\}_{k=1}^N$ across groups for each local dataset $\mathscr{D}_k$, where $a^g$ is the model's performance on a local group at client $k$. We compare the distribution of these group discrepancies based on their median values. Low median values (approaching zero) indicate low group bias and vice versa.

*Utility metrics.* measure overall model performance across clients. Utility is assessed using the average accuracy across all clients.

### 3.3 nonIID Study: Results Overview

This section addresses the impact of data heterogeneity on subgroup fairness across clients in FL. We summarize our findings based on the *subgroup bias, group bias, and utility metrics* in Figure 2.

***Observation:*** TPR of the global model varies across subgroups due to feature distribution heterogeneity, resulting in subgroup bias, as shown in Figure 2 (left) where all subgroups across all datasets have subgroup bias metrics with medians $M$ differing from zero. Figure 2 (right) demonstrates that high model utility still leads to subgroup bias: *high average accuracy does not guarantee any subgroup bias*. This trend is evident from simple to complex datasets (MNIST, FMIST, UTK, FER), illustrating subgroup bias relative to average model performance, with MNIST showing the highest discrepancy. The theoirtical proof of this trend is presented in Equation 11 through Theorem C.1.1, linking subgroup discrepancies to data heterogeneity across clients, quantified by $\Gamma$. MNIST's simplicity and sensitivity to feature distribution variations, such as feature noise, likely contribute to this discrepancy. The data's low complexity makes it sensitive to minor variations, amplifying subgroup performance variations. This motivates us to use Lipschitz-based constraints, which show promise for addressing subgroup bias while preserving model utility. where MNIST's simplicity and sensitivity to heterogeneity significantly amplify performance discrepancies across subgroups due to its low complexity and susceptibility to minor variations.

***Takeaway:*** *Non-IID subgroup data across clients leads to subgroup bias. High-utility techniques may still fall short due to the non-IID nature of subgroups, so addressing this bias is key to improving the fairness and effectiveness of FL systems.*

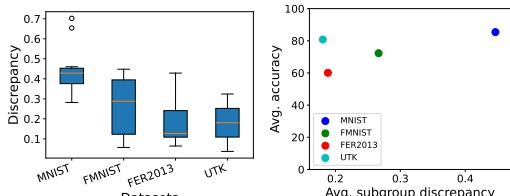

Figure 2: (left) Variation in TPR among subgroups, (right) average model utility across clients.

## 4 LipFed Optimization Framework

### 4.1 Overview

In this section, we formalize a global subgroup fairness constraint for training fair FL models on *individually similar subgroups* $\boldsymbol{X}_k^g$ and $\boldsymbol{X}_{k'}^g$ across different clients $k$ and $k'$, as shown in Figure 3. The Lipschitz property defined in §3 enables this constraint. The global

subgroup fairness constraint $\mathscr{C}_f(\boldsymbol{\theta})$ over each client's local *empirical risk* $R(\boldsymbol{X}_k; \boldsymbol{\theta}_k)$ is defined as:

$$\min_{\boldsymbol{\theta}_k \leftarrow \boldsymbol{\theta}} R(\boldsymbol{X}_k, \boldsymbol{\theta}_k) \quad \text{s.t} \quad \forall \boldsymbol{X}_k^g, \boldsymbol{X}_{k'}^g \in G : \mathscr{C}_f(\boldsymbol{\theta}) = D(h_{\boldsymbol{\theta}}(\boldsymbol{X}_k^g), h_{\boldsymbol{\theta}}(\boldsymbol{X}_{k'}^g)) = \|h_{\boldsymbol{\theta}}(\boldsymbol{X}_k^g) - h_{\boldsymbol{\theta}}(\boldsymbol{X}_{k'}^g)\| \leq d(\boldsymbol{X}_k^g, \boldsymbol{X}_{k'}^g) \tag{4}$$

**Challenges.** Directly using the Lipschitz condition in Equationequation 4 for subgroup fairness in FL poses two challenges:

- *Lack of well-defined similarity metric:* No well-defined metric $d(\cdot)$ exists to to assess the similarity between decentralized subgroups $\boldsymbol{X}_k^g$ and $\boldsymbol{X}_{k'}^g$.
- *Decentralized subgroups:* Unlike centralized machine learning, subgroups $\boldsymbol{X}_k^g$ and $\boldsymbol{X}_{k'}^g$ are spread across clients in FL, making it difficult to assess and impose the Lipschitz condition without breaking FL privacy.

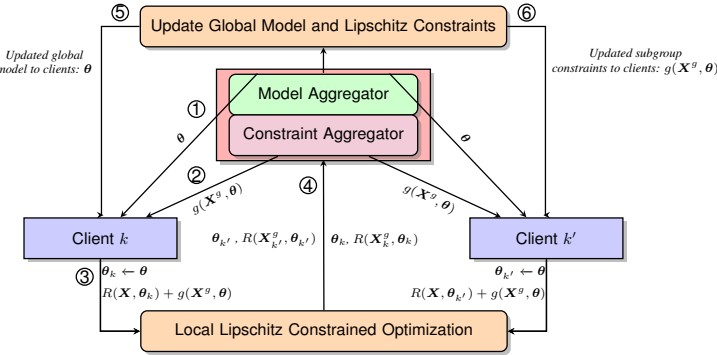

Figure 3: Schematic of our proposed subgroup bias mitigation approach LipFed. $\boldsymbol{X}_k^g$ is the subgroup data and $\boldsymbol{\theta}_k$ are the local model parameters for client $k$. $R(\boldsymbol{X}_k^g, \boldsymbol{\theta}_k)$ and $R(\boldsymbol{X}, \boldsymbol{\theta}_k)$ measure the subgroup and overall data risks, respectively. The numbered circle indicates sequential FL steps.

To solve these challenges, using the subgroup notion that denotes a small set $\boldsymbol{X}_k^g$ with samples that belong to a group $g \in G$, we first use a subgroup similarity metric reflecting the *best* available approximation for assessing similarity between subgroups. This approximation relies on the intuition that subgroups from an individual group have similar characteristics, causing the distance in the similarity metric to be smaller, say $\epsilon$. Computing the subgroup distance across many pairwise subgroup outcomes $D(h_{\boldsymbol{\theta}}(\boldsymbol{X}_k^g), h_{\boldsymbol{\theta}}(\boldsymbol{X}_{k'}^g)) = \|h_{\boldsymbol{\theta}}(\boldsymbol{X}_k^g) - h_{\boldsymbol{\theta}}(\boldsymbol{X}_{k'}^g)\|$ at client $k$ hosting the local subgroup $\boldsymbol{X}_k^g$ poses computation and privacy issues, as there's a lack of global information about decentralized subgroups $\boldsymbol{X}_{k'}^g$ residing on other clients $k'$. To counter that, we compute the subgroup distance across each subgroup outcome and the weighted aggregation of decentralized subgroup outcomes from other clients' $k'$ as:

$$D(h_{\boldsymbol{\theta}}(\boldsymbol{X}_k^g; \boldsymbol{\theta}), h_{\boldsymbol{\theta}}(\boldsymbol{X}_k^g; \boldsymbol{\theta}) = \|R(\boldsymbol{X}_k^g; \boldsymbol{\theta}) - R(\boldsymbol{X}_{k'}^g; \boldsymbol{\theta})\| \approx \|R(\boldsymbol{X}_k^g; \boldsymbol{\theta}) - \sum_{k'} w_{g,k'} R(\boldsymbol{X}_{k'}^g; \boldsymbol{\theta})\| \tag{5}$$

where $w_{g,k'}$ denotes the relative importance of loss weight for client $k'$ in the aggregation. The expression $\|\cdot\|$ quantifies the total discrepancy or distance between the loss performances of the global model on client $k$'s subgroup and the weighted subgroup losses across other clients $k'$. A small discrepancy value indicates that the model's subgroup performance aligns well with all clients' collective performance without bias.

$$\min_{\boldsymbol{\theta}_k \leftarrow \boldsymbol{\theta}} \frac{1}{K} \sum_{k=1}^{K} R(\boldsymbol{X}_k, \boldsymbol{\theta}_k) \quad \text{s.t} \quad \forall \boldsymbol{X}_k^g, \boldsymbol{X}_{k^i}^g \in G : \mathscr{C}_f(\boldsymbol{\theta}) = \sum_{g=1}^{n_g} \|R(\boldsymbol{X}_k^g; \boldsymbol{\theta}) - \sum_{k'} w_{g,k'} R(\boldsymbol{X}_{k'}^g; \boldsymbol{\theta})\| \leq \epsilon \tag{6}$$

The subgroup fairness constraint $\mathscr{C}_f$ of the optimization problem given by 6 ensures that the difference between the loss of a subgroup on client $k$ and the aggregated losses of the same subgroup across other clients $k'$ is small (relative to the upper bound of a slight difference in similar subgroups $\epsilon$), weighted by $w_{g,k}$.

**Importance weights** $w_{g,k}$ **based on subgroup variance** play a pivotal role in LipFed by reflecting each client's contribution to the global model's performance, where higher weights signify greater importance. This insight prompts us to compute importance weights in LipFed that inversely correlate with subgroup variance, ensuring that subgroups with high variance are assigned lower importance in the FL process. We calculate these weights inversely to subgroup variance to ensure subgroups with higher variance in their features (which is known to degrade performance Khani & Liang (2020)) are assigned lesser importance, with weights computed as $w_{g,k} = \frac{1}{AV_{g,k}}$, where $AV_{g,k}$ represents the average feature variance within each subgroup. This approach ensures that subgroups with lower variance receive higher importance weights, thus contributing more effectively to the global model's performance. The detailed formulation is in §B.5.

**Unconstrained problem.** Using the log barrier, we reformulate Equation 6 as an unconstrained problem, which is smooth and differentiable.

$$\min_{\boldsymbol{\theta}_k \longleftarrow \boldsymbol{\theta}} R(\boldsymbol{X}_k, \boldsymbol{\theta}_k) - \frac{1}{t} \log(-(\mathscr{C}_f(\boldsymbol{\theta}) - \epsilon)) \equiv \min_{\boldsymbol{\theta}_k \longleftarrow \boldsymbol{\theta}} R(\boldsymbol{X}_k, \boldsymbol{\theta}_k) - \frac{1}{t} \log(-g(\boldsymbol{X}_g; \boldsymbol{\theta})) \quad (7)$$

where $g(\boldsymbol{X}_g; \boldsymbol{\theta}) = \mathscr{C}_f(\boldsymbol{\theta}) - \epsilon$. This problem minimizes subgroup performance discrepancies across nonIID subgroups while not substantially degrading group fairness by adding a logarithmic barrier to the original objective function. The barrier penalizes constraint violations, creating a "barrier" that prevents the optimizer from straying into infeasible regions of the solution space.

**Computing Optimal** $t$. We use a logarithmic barrier in Equation 7 to achieve subgroup fairness without degrading group fairness significantly. The parameter $t$ controls the barrier's strength; as $t$ increases, the barrier weakens, allowing exploration near the feasible region's boundary. For LipFed, we initialize $t$ at 5 and increase it by $\mu = 1.1$ after each round, following the setup in Kervadec et al. (2019). This strategy relaxes constraints early on to focus on data learning, gradually tightening them as optimization progresses.

### 4.2 THEORETICAL ANALYSIS

This section presents a theoretical analysis of subgroup and group fairness in ML models. Theorems here (proofs provided in §C) establish upper bounds for LipFed optimization and explore trade-offs between Lipschitz continuity, empirical risk, and fairness constraints. These theorems provide insights into the relationships between model properties, fairness constraints, and empirical risk outcomes.

**Theorem 4.2.1.** *Subgroup fairness upper bound. Under* Assumption 1 *in §C.1.1 for any subgroups* $\boldsymbol{X}_k^g$ *and* $\boldsymbol{X}_{k'}^g$ *at clients $k$ and $k'$, we have:*

$$Disc_h(\boldsymbol{X}_k^g, \boldsymbol{X}_{k'}^g) \leq \epsilon^2 \cdot \Gamma \quad \forall g \in \boldsymbol{G}; k, k' \in K : k \neq k'; \epsilon > 0 \quad (8)$$

*where $\Gamma = R(\cdot; \boldsymbol{\theta})^* - \sum_{k=1}^{K} p_k R_k(\cdot; \boldsymbol{\theta}_k)^*$ quantifies the degree of data heterogeneity; if the data are non-iid, then $\Gamma$ is nonzero and its magnitude reflects the heterogeneity of the data distribution Li et al. (2019b). $p_k$ is the weight of the $k$-th device such that $p_k$ is proportional the device's local data size and $p_k \geq 0$.*

**Theorem 4.2.2.** *Group fairness upper bound. Under Assumption 1-5 in §C.1.2 on the global empirical risk function $R(\boldsymbol{X}; \boldsymbol{\theta})$ as per recent FL works Li et al. (2019a;b), we have:*

$$Disc_h(\boldsymbol{X}_k^g, \boldsymbol{X}_k^{g'}) \leq \frac{\kappa}{\gamma + \Gamma - 1} \cdot \left( \frac{2B}{\mu} + \epsilon^2 \cdot \Gamma \right) \quad \forall g, g' \in \boldsymbol{G}; g \neq g' \quad (9)$$

*where $\Gamma$ is as defined above, $\kappa = \frac{L}{\mu}$, $B = \sum_{k=1}^{K} p_k^2 \sigma_k^2 + 6L\Gamma + 8(E-1)^2 G^2$, $E$ is the number of local training rounds/epochs for each device $k$, and $\gamma = \max\{8\kappa, E\}$.*

#### 4.2.1 UNDERSTANDING THE RELATIONSHIP BETWEEN SUBGROUP AND GROUP BIAS MITIGATION

To understand the unexplored interplay between subgroup and group fairness, we examine how changing the common bounds parameter $\epsilon$ affects both subgroup and group fairness. We clarify the distinction between subgroup and group fairness interpretations in centralized versus decentralized settings in §B.

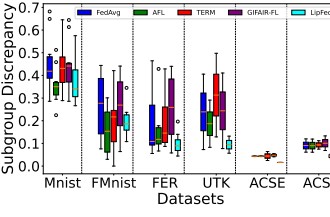

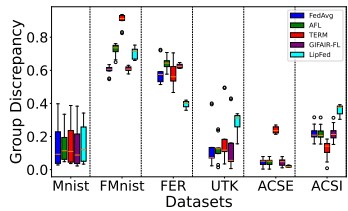

(a) TPR variations in subgroups.  (b) TPR variations in groups.

Figure 4: Demonstrating subgroup bias in model performance for different datasets and baselines.

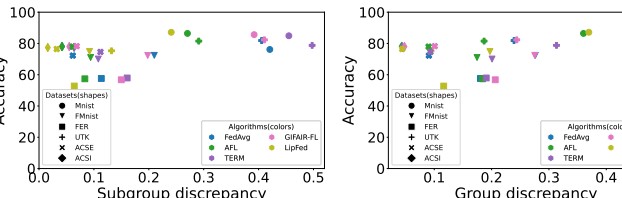

(a) Utility vs. Subgroup fairness.  (b) Utility vs. Group fairness.

Figure 5: Demonstrating model utility vs. discrepancy for different datasets and baselines.

The interplay between subgroup and group fairness is critical for achieving equitable outcomes across both demographic groups and subgroups. Subgroup fairness aims to ensure fair treatment by reducing the parameter $\epsilon$, which in turn minimizes $\epsilon^2 \cdot \Gamma$, as detailed in Theorem 4.2.1. While lowering $\epsilon$ enhances subgroup fairness, it can have mixed effects on group fairness. Group fairness, on the other hand, is governed by the bounds in Theorem 4.2.2, which include terms such as $\frac{\kappa}{\gamma + \Gamma - 1}$ and $\frac{2B}{\mu}$. Although reducing $\epsilon$ may initially lower the group fairness bound through the term $\epsilon^2 \cdot \Gamma$, larger values of $\kappa$, $B$, or $\gamma$ may overshadow these benefits, highlighting the need for careful calibration of $\epsilon$ to balance both dimensions of fairness effectively.

## 5 EXPERIMENTS

In this section, we evaluate LipFed's effectiveness in mitigating subgroup bias to assess whether LipFed achieves subgroup fairness across diverse clients while adhering to three key constraints: (1) maintaining group fairness; (2) preserving model utility and (3) data privacy.

### 5.1 EXPERIMENTAL SETUP

**Models and datasets.** Our study assesses LipFed's efficacy using the setup in §3.2. We compare LipFed with SOTA baselines on benchmark datasets and evaluate its real-world applicability using the UTK dataset and ACS fairness dataset, examining bias mitigation across different client partitions in FL.

**Baselines.** We evaluate LipFed across two key categories, scrutinizing bias reduction, model utility, privacy and group fairness tradeoff. 1) The *FL baseline category* represented by FedAvg, serves as the standard learning scheme in FL. 2) The *FL bias-reduction category* includes AFLMohri et al. (2019), TERMLi et al. (2020), and GIFAIR-FL Yue et al. (2021), which use empirical risk reweighting to mitigate bias and adapt the global model to diverse local data distributions (*Note: We use client and group bias baselines, as to the best of our knowledge, no existing techniques are specifically designed to address subgroup bias. We provide additional evaluation of FL robustness techniques that are not specifically focused on fairness in §G.1* ).

### 5.2 COMPARATIVE EVALUATION OF LIPFED ON BENCHMARK AND REAL-WORLD DATASETS

We use six datasets to compare LipFed with bias mitigation baselines in achieving subgroup fairness. In the MNIST and Fashion-MNIST datasets, LipFed significantly outperforms the baselines in reducing subgroup

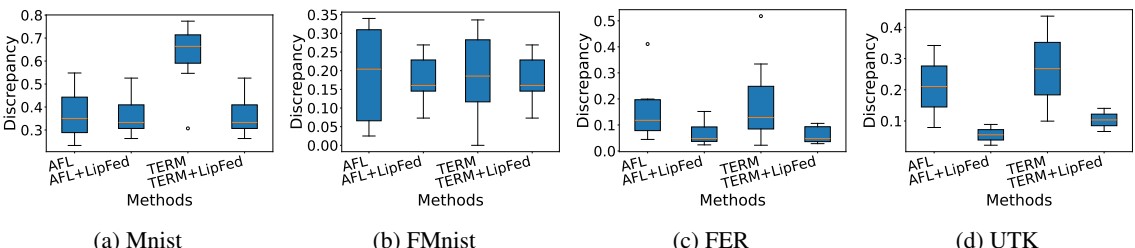

(a) Mnist        (b) FMnist        (c) FER        (d) UTK

Figure 6: Enhancing fairness across other FL algorithms: LipFed Elevates traditional FL algorithms in subgroup bias mitigation across datasets.

bias, as illustrated in Figure 4a. This improvement is largely due to LipFed's use of Lipschitz continuity constraints, which directly address discrepancies in subgroup performance. In contrast, existing fairness techniques focus primarily on group fairness, which does not inherently guarantee subgroup fairness. However, LipFed occasionally exhibits higher median group discrepancies ( Figure 4b), indicating that improving subgroup fairness does not always translate into improved group fairness, a point further explored in the theoretical analysis §4.2. Nevertheless, LipFed maintains competitive model utility compared to baseline methods not only at the subgroup level ( Figure 5a) but also at the group level ( Figure 5b). The trends are consistent in real-world datasets (FER2013, UTK, ACSI, and ACSE) with those observed in the benchmark datasets, validating LipFed's ability to balance subgroup fairness and utility in practical, non-IID FL settings.

*Takeaway: LipFed mitigates subgroup bias for non-IID subgroups across clients and maintains competitive utility compared to baselines without compromising performance on all six datasets.*

### 5.3 IMPACT OF LIPFED INTEGRATION WITH TRADITIONAL FL METHODS ON SUBGROUP FAIRNESS

We evaluate the impact of combining LipFed with other FL algorithms, such as AFL and TERM, to reduce subgroup bias. Our goal is to *determine whether LipFed can address subgroup fairness beyond the FedAvg technique, particularly in scenarios with feature heterogeneity*. By integrating LipFed with AFL and TERM, resulting in AFL+LipFed and TERM+LipFed, we aim to ensure consistent model performance across clients. Using the same datasets and metrics, we find that both AFL+LipFed and TERM+LipFed consistently demonstrate lower median subgroup discrepancies compared to AFL and TERM alone ( Figure 6). This improvement is driven by LipFed's enforcement of Lipschitz continuity constraints, which specifically target and penalize subgroup performance discrepancies. In contrast, most fairness techniques focus primarily on group fairness, which is insufficient to fully address subgroup fairness challenges.

*Takeaway. LipFed enhances effectiveness of other group fairness methods in FL, in reducing subgroup bias.*

### 5.4 TRADE-OFF BETWEEN SUBGROUP AND GROUP FAIRNESS

Figure 7 illustrates the empirical trade-off between subgroup and group fairness, complementing the theoretical analysis discussed earlier. The red lines indicate trends in various algorithms' ability to mitigate subgroup and group bias. A negative slope highlights the trade-off, where improving one type of fairness often compromises the other. LipFed, shown at the leftmost marker, effectively enhances subgroup fairness but slightly compromises group fairness due to the challenge of balancing these trade-offs during optimization. The mixed trends observed can be attributed to *Dataset characteristics and feature distribution* as they influence this trade-off. For instance, MNIST's uniform feature distribution helps align subgroup and group fairness, whereas FMNIST's variability in textures and styles causes a divergence between the two. Our results show that bias mitigation techniques exhibit varying trends depending on factors like data heterogeneity and training parameters (e.g., $\epsilon$). Careful parameter tuning is key to balancing subgroup and group fairness, with dataset complexity playing a major role in their alignment or divergence across clients.

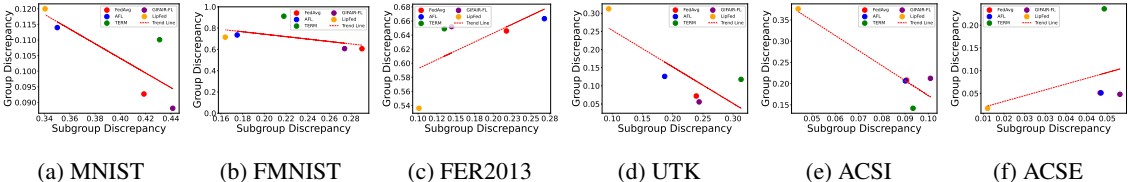

|       (a) MNIST       |       (b) FMNIST       |       (c) FER2013       |       (d) UTK       |       (e) ACSI       |       (f) ACSE       |

Figure 7: Group Fairness vs. Subgroup Fairness on different baselines and datasets.

*Takeaway: Balancing subgroup and group fairness requires trade-offs and careful parameter tuning.*

### 5.5 PRIVACY PRESERVATION AND ITS IMPACT ON FAIRNESS AND UTILITY

To assess the impact of differential privacy on subgroup fairness and model performance, we introduce varying levels of Laplace noise, with $\epsilon \in \{0.8, 1.0, 1.4\}$, to the local subgroup losses exchanged between clients and the server. This technique ensures that sensitive client metadata remains protected while allowing for calculating fairness constraints. The $\epsilon$ values range aligns with standard privacy-preserving practices in FL Abay et al. (2020).

We evaluate the impact of different privacy levels on subgroup discrepancy and model accuracy for benchmark datasets. As shown inTable 1, differential privacy has minimal effect on subgroup fairness and utility. For instance, at $\epsilon = 0.8$, MNIST shows a discrepancy of 0.25 and 87.11% accuracy, while Fashion-MNIST shows a 0.1 discrepancy and 74.5% accuracy. These results remain consistent across varying privacy levels and without privacy (no-DP), indicating that privacy does not significantly degrade fairness or performance.

Table 1: Impact of differential privacy levels on subgroup fairness and model utility.

| $\epsilon$ | MNIST | | Fashion-MNIST | |
|---|---|---|---|---|
| | Sub. Disc. | Avg. Acc. | Sub. Disc. | Avg. Acc. |
| 0.8 | 0.25 | 87.11% | 0.1 | 74.54% |
| 1.0 | 0.24 | 87.11% | 0.1 | 74.54% |
| 1.4 | 0.24 | 87.12% | 0.09 | 74.54% |
| no-DP | 0.24 | 87.13% | 0.09 | 74.55% |

LipFed's inherent Lipschitz continuity and subgroup similarity provide natural privacy protection by reducing sensitivity to individual data points, without needing explicit noise addition. The mathematical framework in §C.4 can be used to argue that our technique naturally satisfies differential privacy criteria, meaning the technique limits information leakage about individual data points in the dataset to the extent that no single data point significantly alters the statistical characteristics of the output, thereby offering privacy protection as an inherent feature.

***Takeaway.*** *LipFed effectively preserves sensitive client information through differential privacy while having only a negligible impact (0.01%) on model accuracy and maintaining stable subgroup fairness.*

## 6 CONCLUSION

The heterogeneity of statistical features in local data across clients in FL models leads to subgroup bias. To address this, we introduce LipFed, a framework leveraging the Lipschitz fairness constraint LipFed ensures that similar subgroups have performance outcomes with a statistical distance within their similarity measure, improving subgroup fairness without significantly sacrificing utility, as delineated by our theoretical analysis which shows a trade-off in group fairness. Our extensive experiments validate LipFed's efficacy in subgroup bias mitigation, demonstrating its superiority over six state-of-the-art bias mitigation techniques and enhancing the fairness of traditional FL methods.

## 7 REPRODUCIBILITY STATEMENT

We outline the reproducibility of our work on mitigating subgroup bias in FL through comprehensive documentation and resource sharing. LipFed, is detailed in Section 4 of the main text, where we outline the algorithmic framework and its theoretical underpinnings. The assumptions leading to our theoretical results are specified in Section 4.2, alongside complete proofs of the claims in Appendix C.

For reproducibility of the experimental results, we provide a thorough description of the datasets utilized, including benchmark and real-world datasets, in Appendix E. The specific data processing steps and partitioning methodologies are outlined in the experimental setup section and Appendix E.

To facilitate ease of reproduction, we provide an anonymous link to our source code and the scripts used for our experiments in the supplementary materials in Appendix E.2. This code includes implementations of the LipFed algorithm and details on the parameter settings for all experiments conducted. We believe that these resources, combined with the clear delineation of methods and assumptions within the paper, will assist researchers in reproducing our results accurately.

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

# Appendix

We provide additional information for our paper, *LipFed: Mitigating Subgroup Bias in Federated Learning with Lipschitz Constraints*, in the following order:

- Limitations and Future Work (Appendix A)
- Terminology/Techniques (Appendix B)
- Additional Analysis (Appendix C)
- Experimental Setup (Appendix E)
- Metrics (Appendix F)
- Additional Results (Appendix G

## A    LIMITATIONS AND FUTURE WORK

### A.1    LIMITATIONS

Despite the effectiveness of the LipFed framework in mitigating subgroup bias, several limitations remain. Firstly, the reliance on the Lipschitz property to ensure subgroup fairness introduces constraints that may not universally apply across all types of models or datasets. There is a possibility that different models exhibit varying degrees of sensitivity to Lipschitz constraints, which could lead to inconsistent results when applied to non-IID data distributions. Second, the effectiveness of our method is influenced by the proper selection of the hyperparameter $\epsilon$ that governs the Lipschitz constraint. Finding the optimal balance between subgroup and group fairness may require extensive tuning and could differ based on the specific characteristics of the datasets being used.

Furthermore, while our approach shows improvements over existing methods, the trade-off between subgroup and group fairness necessitates careful calibration, which may not be straightforward. As subgroup variance decreases, the potential for bias to still emerge in certain groups remains a challenge. Lastly, the additional computational overhead of enforcing Lipschitz constraints during the optimization process may not be feasible for all practical applications, especially in resource-constrained environments.

### A.2    FUTURE WORK

Further empirical studies are needed to evaluate the performance of LipFed in diverse real-world scenarios, including applications beyond image classification, such as text and audio data. Investigating the scalability of our method in federated learning environments with a large number of clients and significantly diverse data distributions would also be beneficial.

Moreover, it would be valuable to explore dynamic tuning mechanisms for the hyperparameter $\epsilon$, potentially through adaptive methods that can adjust to the evolving characteristics of the data during the training process. This would facilitate achieving a more nuanced balance between subgroup and group fairness.

## B    TERMINOLOGY/TECHNIQUES

### B.1    NOVELTY OF LIPSCHITZ CONSTRAINTS

While Lipschitz continuity itself is not a novel concept, our work introduces one of the first adaptations of Lipschitz constraints in FL to specifically address subgroup fairness. LipFed leverages these constraints to

calculate the importance of each subgroup on a client, enabling the model to assign different weights to subgroups based on the variability in their data. This approach helps mitigate the effects of non-IID data by prioritizing subgroups that experience greater bias.

What sets LipFed apart is its ability to enforce Lipschitz constraints without requiring access to clients' raw data, preserving privacy—a crucial aspect in federated settings. By focusing on the balance between subgroup fairness and data privacy, LipFed offers an innovative solution to address fairness in FL systems without compromising privacy.

### B.2 Federated Learning Subgroup Fairness vs. Centralized Learning Subgroup Fairness

Subgroup fairness in FL differs significantly from centralized learning. In centralized learning, all data is aggregated in one location, making it easier to apply fairness constraints uniformly across subgroups. However, FL operates on decentralized data distributed across multiple clients, where non-IID data distributions pose significant challenges. Achieving subgroup fairness in FL requires ensuring that each client contributes equitably to the global model despite these variations. This decentralized setup demands sophisticated model aggregation techniques to maintain subgroup fairness, as direct access to all client data is not possible.

### B.3 Subgroup Fairness vs. Fairness across multiple sensitive attributes

Fairness across multiple sensitive attributes, discussed in MultiFairTian et al. (2024), ensures that fairness constraints are satisfied *for each sensitive attribute individually* (regardless of their number) without necessarily focusing on their intersections. Consider a loan approval algorithm that aims to ensure fairness. The algorithm might be designed to approve loans at the same rate for men and women (gender fairness) and at the same rate for people of different ages (age fairness). Each attribute (gender, age) is treated separately to ensure fairness, but the algorithm might not specifically check if it's fair to, for instance, young women or older men. Subgroup fairness (intersectional attributes focus) and multiple sensitive attributes (individual attribute focus) have some overlap, but they are not closely related. The distinction between these approaches is well-recognized in the literature Kearns et al. (2018). In centralized learning, there is a clear separation between ensuring fairness for individual attributes and addressing fairness at the intersection of multiple attributes (subgroup fairness). As noted in the paper Kearns et al. (2018), the need to ensure fairness across intersectional subgroups is paramount to avoid fairness gerrymandering, where a model appears fair across individual attributes but fails at the intersection of these attributes.

### B.4 Additional Causes for Subgroup Fairness

Several factors contribute to subgroup unfairness, one of the most prominent being differences in group sizes. This issue is commonly referred to as *label distribution skew*, where imbalances in the distribution of labels across groups lead to biased outcomes. This challenge has been extensively studied in recent federated learning fairness research Yue et al. (2023); Kearns et al. (2018).

In contrast, our work Lipfed deliberately focuses on a less explored yet equally important issue: the *same label, different features* phenomenon. This refers to instances where subgroups that share the same label exhibit significantly different feature distributions, leading to unfair treatment across those subgroups. By addressing this underexamined factor, our work provides new insights into the complexities of achieving subgroup fairness in FL.

## B.5 Average Variance of Image Pixel Weighting Scheme

Pixel-level variance reflects differences in texture, lighting, and other visual features that affect image data similarity and heterogeneity Zhang & LeCun (2015). By computing subgroup importance weights based on the average variance of image pixels, subgroups with higher pixel variance, indicating less robustness, are prioritized during training to improve model performance Wang et al. (2004). In Khani & Liang (2020), the authors present a mathematical framework showing how feature variance, such as image pixel variance, influences fairness by affecting loss discrepancy. Here are the relevant equations and their implications in scenarios of binary groups (0 and 1, say):

$$Disc \propto \left| (\Lambda\beta)^\top \Delta\Sigma_z (\Lambda\beta) - (P[g=1] - P[g=0])((\Lambda\beta)^\top \Delta\mu_z)^2 \right| \tag{10}$$

where $\Lambda = (\Sigma_z + \Sigma_u)^{-1}\Sigma_u$ is a matrix that balances the variance of the latent features ($\Sigma_z$) with the variance of noise in those features ($\Sigma_u$), ensuring that features with lower noise are weighted more heavily.

The terms $\Delta\Sigma_z = Var[z \mid g=1] - Var[z \mid g=0]$ and $\Delta\mu_z = E[z \mid g=1] - E[z \mid g=0]$ represent the difference in the variance and the mean of the latent features between the two groups, $g=1$ and $g=0$, respectively. Larger differences in these values signify a greater potential for bias, as one group's feature distribution deviates significantly from the other's. The proportions $P[g=1]$ and $P[g=0]$ reflect the relative sizes of the two groups, which influence how much weight the second term in the equation has on the overall discrepancy.

The model's learned parameters, $\beta$, determine the importance of each latent feature in the prediction process. The interaction between the feature variances and the model parameters, captured by the term $(\Lambda\beta)^\top \Delta\Sigma_z (\Lambda\beta)$, increases as feature variance ($\Sigma_z$) increases, indicating that higher variance in features leads to a larger loss discrepancy between groups.

Building on previous studies, we assign higher importance to subgroups with higher variance, which indicates potential model bias. This method aligns with other techniques that prioritize training samples based on characteristics like gradient norm, assessing robustness through feature heterogeneity. This loss discrepancy directly contributes to model bias, as it suggests unequal treatment of different groups. Our weighting scheme aims to mitigate this bias by assigning higher importance to subgroups with greater variance. We compare our fairness weighting scheme with GIFAIR-FL, a framework for fairness in FLYue et al. (2023). GIFAIR-FL uses regularization to penalize variations in client group losses, adapting to statistical differences at each communication round. This approach aligns with our fairness definitions by ensuring equitable performance across data groups.

## C Additional Analysis

### C.1 Theoretical Analysis

This section presents a theoretical analysis of subgroup and group fairness in machine learning models. The theorems discussed here aim to establish upper bounds and explore trade-offs between Lipschitz continuity, empirical risk, and fairness constraints. Theorem 3.4.1 addresses the upper bound for subgroup fairness under Lipschitz continuity conditions, providing insights into the absolute difference in empirical risk between subgroups. Moving forward, Theorem 3.4.2 extends this analysis to group fairness, establishing upper bounds based on smoothness properties. Finally, Theorem 3.4.3 delves into the trade-off analysis between Lipschitz constraints and empirical risk performance, shedding light on how tighter fairness constraints can impact model adaptability and the overall expected discrepancy in empirical risk across different groups. These theorems collectively contribute to understanding the intricate relationship between model properties, fairness constraints, and empirical risk outcomes.

**Theorem C.1.1.** *Subgroup Fairness Upper Bound. Assumption 1.* $R(\cdot; \boldsymbol{\theta})$ *is* $(D, d)$*-Lipschitz continuous (since it was enforced during optimization).*

*Then, for any subgroups* $\boldsymbol{X}_k^g$ *and* $\boldsymbol{X}_{k'}^g$ *at clients* $k$ *and* $k'$ *respectively, we have:*

$$Disc_h(\boldsymbol{X}_k^g, \boldsymbol{X}_{k'}^g) \leq \epsilon^2 \cdot \Gamma \quad \forall g \in \boldsymbol{G}; k, k' \in K : k \neq k'; \epsilon > 0 \tag{11}$$

*where* $\Gamma = R(\cdot; \boldsymbol{\theta})^* - \sum_{k=1}^{K} p_k R_k(\cdot; \boldsymbol{\theta}_k)^*$ *(* $R^*$ *and* $R_k^*$ *are the minimum values of* $R^*$ *and* $R_k^*$*, respectively) quantifies the degree of data heterogeneity; If the data are non-iid, then* $\Gamma$ *is nonzero, and its magnitude reflects the heterogeneity of the data distribution Li et al. (2019b).*

**Proof:** *We start with the Lipschitz continuity property for predictions:*

$$D(h_{\boldsymbol{\theta}}(\boldsymbol{X}_k^g; \boldsymbol{\theta}), h_{\boldsymbol{\theta}}(\boldsymbol{X}_{k'}^g; \boldsymbol{\theta})) \leq \epsilon \cdot d(\boldsymbol{X}_k^g, \boldsymbol{X}_{k'}^g) \quad \forall g \in \boldsymbol{G}; k, k' \in K : k \neq k' \tag{12}$$

*This inequality tells us that the distance between predictions made by the model* $h_{\boldsymbol{\theta}}(\boldsymbol{X}_k^g; \boldsymbol{\theta})$ *and* $h_{\boldsymbol{\theta}}(\boldsymbol{X}_{k'}^g; \boldsymbol{\theta})$ *is bounded by the Lipschitz constant* $\epsilon$ *times the distance between the subgroups* $\boldsymbol{X}_k^g$ *and* $\boldsymbol{X}_{k'}^g$*.*

*The absolute difference in the subgroup risk functions due to different predictions can be expressed as:*

$$|R(\boldsymbol{X}_k^g; \boldsymbol{\theta}) - R(\boldsymbol{X}_{k'}^g; \boldsymbol{\theta})| = |f(h_{\boldsymbol{\theta}}(\boldsymbol{X}_k^g; \boldsymbol{\theta})) - f(h_{\boldsymbol{\theta}}(\boldsymbol{X}_{k'}^g; \boldsymbol{\theta}))| \quad \forall g \in \boldsymbol{G}; k, k' \in K : k \neq k' \tag{13}$$

*Here,* $f$ *is a function that maps predictions to risk values.*

*Now, we substitute the Lipschitz continuity property for predictions into the risk function equation:*

$$|f(h_{\boldsymbol{\theta}}(\boldsymbol{X}_k^g; \boldsymbol{\theta})) - f(h_{\boldsymbol{\theta}}(\boldsymbol{X}_{k'}^g; \boldsymbol{\theta}))| \leq \epsilon \cdot D(h_{\boldsymbol{\theta}}(\boldsymbol{X}_k^g; \boldsymbol{\theta}), h_{\boldsymbol{\theta}}(X_{k'}^g; \boldsymbol{\theta})) \quad \forall g \in \boldsymbol{G}; k, k' \in K : k \neq k' \tag{14}$$

*Since we know that* $D(h_{\boldsymbol{\theta}}(\boldsymbol{X}_k^g; \boldsymbol{\theta}), h_{\boldsymbol{\theta}}(X_{k'}^g; \boldsymbol{\theta}))$ *is bounded by* $\epsilon \cdot d(\boldsymbol{X}_k^g, \boldsymbol{X}_{k'}^g)$*, we can replace it in the inequality above. This substitution leads to the following inequality:*

$$|f(h_{\boldsymbol{\theta}}(\boldsymbol{X}_k^g; \boldsymbol{\theta})) - f(h_{\boldsymbol{\theta}}(\boldsymbol{X}_{k'}^g; \boldsymbol{\theta}))| \leq \epsilon \cdot (\epsilon \cdot d(\boldsymbol{X}_k^g, \boldsymbol{X}_{k'}^g)) = \epsilon^2 \cdot d(\boldsymbol{X}_k^g, \boldsymbol{X}_{k'}^g) \quad \forall g \in \boldsymbol{G}; k, k' \in K : k \neq k' \tag{15}$$

$$\implies \max\{f(h_{\boldsymbol{\theta}}(\boldsymbol{X}_k^g; \boldsymbol{\theta})) - f(h_{\boldsymbol{\theta}}(\boldsymbol{X}_{k'}^g; \boldsymbol{\theta}))\} \leq \epsilon^2 \cdot d(\boldsymbol{X}_k^g, \boldsymbol{X}_{k'}^g) \quad \forall g \in \boldsymbol{G} \tag{16}$$

$$\implies \max\{R(\boldsymbol{X}_k^g; \boldsymbol{\theta}) - R(\boldsymbol{X}_{k'}^g; \boldsymbol{\theta})\} \leq \epsilon^2 \cdot \Gamma \tag{17}$$

$$\implies Disc_h(\boldsymbol{X}_k^g, \boldsymbol{X}_{k'}^g) \leq \epsilon^2 \cdot \Gamma \tag{18}$$

**Theorem C.1.2.** *Group fairness upper bound. Suppose that the following assumptions hold on the global empirical risk function* $R(\boldsymbol{X}; \boldsymbol{\theta})$ *according to recent works in FL Li et al. (2019a;b),*

**Assumption 1.** $R_1, \ldots, R_K$ *are all L-smooth: for all* $\boldsymbol{\theta}_1$ *and* $\boldsymbol{\theta}_2$*,*

$$R_k(\boldsymbol{\theta}_1) \leq R_k(\boldsymbol{\theta}_2) + (\boldsymbol{\theta}_1 - \boldsymbol{\theta}_2)^T \nabla R_k(\boldsymbol{\theta}_2) + \frac{L}{2} \|\boldsymbol{\theta}_1 - \boldsymbol{\theta}_2\|_2^2.$$

**Assumption 2.** $R_1, \ldots, R_K$ *are all* $\mu$*-strongly convex: for all* $\boldsymbol{\theta}_1$ *and* $\boldsymbol{\theta}_2$*,*

$$R_k(\boldsymbol{\theta}_1) \geq R_k(\boldsymbol{\theta}_2) + (\boldsymbol{\theta}_1 - \boldsymbol{\theta}_2)^T \nabla R_k(\boldsymbol{\theta}_2) + \frac{\mu}{2} \|\boldsymbol{\theta}_1 - \boldsymbol{\theta}_2\|_2^2.$$

**Assumption 3.** *Let $\xi_k^t$ be sampled from the $k$-th device's local data uniformly at random. The variance of stochastic gradients in each device is bounded:*

$$\mathbb{E}\left[\|\nabla R_k(\boldsymbol{\theta}_k^t, \xi_k^t) - \nabla R_k(\boldsymbol{\theta}_k^t)\|^2\right] \leq \sigma_k^2 \quad for\ k = 1, \ldots, K.$$

**Assumption 4.** *$R(\cdot; \boldsymbol{\theta})$ is $(D, d)$-Lipschitz continuous (since it was enforced during optimization).*

**Assumption 5.** *The expected squared norm of stochastic gradients is uniformly bounded, i.e.,*

$$\mathbb{E}\left[\|\nabla R_k(\boldsymbol{\theta}_k^t, \xi_k^t)\|^2\right] \leq G^2 \quad for\ all\ k = 1, \ldots, K\ and\ t = 1, \ldots, T-1.$$

*Then,*

$$Disc_h(\boldsymbol{X}_k^g, \boldsymbol{X}_k^{g'}) \leq \frac{\kappa}{\gamma + \Gamma - 1} \cdot \left(\frac{2B}{\mu} + \epsilon^2 \cdot \Gamma\right) \quad \forall g, g' \in \boldsymbol{G}; g \neq g' \tag{19}$$

*where $\Gamma = R(\cdot; \boldsymbol{\theta})^* - \sum_{k=1}^K p_k R_k(\cdot; \boldsymbol{\theta}_k)^*$ ($R^*$ and $R_k^*$ are the minimum values of $R^*$ and $R_k^*$, respectively) quantifies the degree of data heterogeneity; If the data are non-iid, then $\Gamma$ is nonzero, and its magnitude reflects the heterogeneity of the data distribution, $\kappa = \frac{L}{\mu}$, $B = \sum_{k=1}^K p_k^2 \sigma_k^2 + 6L\Gamma + 8(E-1)^2 G^2$, $E$ is the number of local training rounds/epochs for each device $k$, and $\gamma = \max\{8\kappa, E\}$.*

**Proof:** *According to Li et al. (2019b), we know that:*

$$\mathbb{E}[R(\cdot; \boldsymbol{\theta}_T)] - R(\cdot; \boldsymbol{\theta})^* \leq \frac{\kappa}{\gamma + \Gamma - 1} \cdot \left(\frac{2B}{\mu} + \frac{\mu\gamma^2}{2}\mathbb{E}_k\|\boldsymbol{\theta} - \boldsymbol{\theta}_k^*\|^2\right) \tag{20}$$

$$\implies \mathbb{E}[R(\boldsymbol{X}_k^g; \boldsymbol{\theta}_T)] - R(\boldsymbol{X}_k^{g'}; \boldsymbol{\theta})^* \leq \max\{\mathbb{E}[R(\boldsymbol{X}_k^g; \boldsymbol{\theta}_T)] - R(\boldsymbol{X}_k^{g'}; \boldsymbol{\theta})^*\}$$
$$\leq \frac{\kappa}{\gamma + \Gamma - 1} \cdot \left(\frac{2B}{\mu} + \frac{\mu\gamma^2}{2}\mathbb{E}_k\|\boldsymbol{\theta} - \boldsymbol{\theta}_k^*\|^2\right) \tag{21}$$

*But, $\|\boldsymbol{\theta} - \boldsymbol{\theta}^*\|^2 \approx D(h_{\boldsymbol{\theta}}(\boldsymbol{X}_k^g), h_{\boldsymbol{\theta}}(\boldsymbol{X}_k^{g'}))$*

$$\therefore \max\{\mathbb{E}[R(\boldsymbol{X}_k^g; \boldsymbol{\theta}_T)] - R(\boldsymbol{X}_k^{g'}; \boldsymbol{\theta})^*\} \leq \frac{\kappa}{\gamma + \Gamma - 1} \cdot \left(\frac{2B}{\mu} + \epsilon \cdot D(h_{\boldsymbol{\theta}}(\boldsymbol{X}_k^g), h_{\boldsymbol{\theta}}(\boldsymbol{X}_k^{g'}))\right); \epsilon = \frac{\mu\gamma^2}{2} \tag{22}$$

$$\implies \max\{\mathbb{E}[R(\boldsymbol{X}_k^g; \boldsymbol{\theta}_T)] - R(\boldsymbol{X}_k^{g'}; \boldsymbol{\theta})^*\} \leq \frac{\kappa}{\gamma + \Gamma - 1} \cdot \left(\frac{2B}{\mu} + \epsilon \cdot (\epsilon \cdot d(\boldsymbol{X}_k^g, \boldsymbol{X}_k^{g'}))\right) \tag{23}$$

$$\implies \max\{\mathbb{E}[R(\boldsymbol{X}_k^g; \boldsymbol{\theta}_T)] - R(\boldsymbol{X}_k^{g'}; \boldsymbol{\theta})^*\} \leq \frac{\kappa}{\gamma + \Gamma - 1} \cdot \left(\frac{2B}{\mu} + \epsilon^2 \cdot \Gamma\right) \tag{24}$$

$$\implies Disc_h(\boldsymbol{X}_k^g, \boldsymbol{X}_k^{g'}) \leq \frac{\kappa}{\gamma + \Gamma - 1} \cdot \left(\frac{2B}{\mu} + \epsilon^2 \cdot \Gamma\right) \tag{25}$$

## C.2 TRADEOFF ANALYSIS

To understand the trade-off between subgroup and group fairness, we examine how changing the common bounds parameter $\epsilon$ affects both subgroup and group fairness:

Improving subgroup fairness is essential to ensure equitable outcomes across different demographic groups. The primary objective in this context is to decrease $\epsilon$ to reduce the term $\epsilon^2 \cdot \Gamma$. This reduction has a direct effect on subgroup fairness by minimizing subgroup discrepancies, as indicated by the relationship in Theorem C.1.1. Regarding group fairness, the decrease in $\epsilon^2 \cdot \Gamma$ contributes to lowering the group fairness bound. However, the overall impact on group fairness is also dependant upon other factors, such as the terms $\frac{\kappa}{\gamma+\Gamma-1}$ and $\frac{2B}{\mu}$. When $\frac{2B}{\mu}$ is substantially large, it might overshadow the benefits gained from reducing $\epsilon$, as this term can dominate the fairness bound.

In improving group fairness, it is crucial to consider the influence of all terms within the group fairness bounds. The group fairness bound is affected by the bound in Theorem C.1.2. Large values of $\kappa$, $B$, or $\gamma$ can significantly impact this bound. Furthermore, adjustments aimed at improving group fairness can have implications for subgroup fairness. Specifically, increasing $\epsilon$ might be necessary to prevent an excessive rise in the group fairness bound. However, this increment will directly raise $\epsilon^2 \cdot \Gamma$, resulting in a higher discrepancy among subgroups. Balancing these factors is crucial for achieving both group and subgroup fairness effectively.

## C.3 BALANCING THE TRADE-OFF

To balance subgroup and group fairness, we need to carefully tune $\epsilon$ while considering the impact of the other parameters. Decreasing $\epsilon$ can lead to improvements in subgroup fairness, as indicated by the reduction in $\epsilon^2 \cdot \Gamma$. This directly minimizes subgroup discrepancies. In terms of group fairness, a decrease in $\epsilon^2 \cdot \Gamma$ can also lead to improvements, particularly if this term is significant within the fairness bound. However, if the term $\frac{2B}{\mu}$ is large, the overall improvement in group fairness may be limited, as this dominant term can overshadow the effects of reducing $\epsilon$.

On the other hand, increasing $\epsilon$ can have adverse effects on subgroup fairness since $\epsilon^2 \cdot \Gamma$ will increase, leading to greater subgroup discrepancies. In terms of group fairness, an increase in $\epsilon$ can potentially yield improvements if the other terms, such as $\frac{\kappa}{\gamma+\Gamma-1}$ and $\frac{2B}{\mu}$, dominate the fairness bound. However, this benefit is constrained if $\epsilon^2 \cdot \Gamma$ is already a small component within the bound.

Balancing these factors is crucial. It involves a trade-off between minimizing subgroup discrepancies and optimizing group fairness, considering the relative magnitudes of the different terms in the fairness bound. Careful tuning of $\epsilon$ is essential to achieve a desirable balance that promotes both subgroup and group fairness.

When the parameters $\kappa$, $B$, or $\gamma$ are large, the group fairness bound is dominated by $\frac{\kappa}{\gamma+\Gamma-1}\left(\frac{2B}{\mu}\right)$, making it less sensitive to changes in $\epsilon$. Increasing $\epsilon$ to maintain group fairness will significantly worsen subgroup fairness. Conversely, when the parameters $\kappa$, $B$, or $\gamma$ are small, the group fairness bound becomes more sensitive to $\epsilon$. In this scenario, decreasing $\epsilon$ to improve subgroup fairness will have a noticeable impact on the group fairness bound. This can potentially compromise group fairness if $\epsilon$ becomes too small.

Balancing subgroup and group fairness requires carefully tuning $\epsilon$ while considering the impact of these other parameters. Decreasing $\epsilon$ can lead to improvements in subgroup fairness, as indicated by the reduction in $\epsilon^2 \cdot \Gamma$, which directly minimizes subgroup discrepancies. In terms of group fairness, a decrease in $\epsilon^2 \cdot \Gamma$ can also lead to improvements, particularly if this term is significant within the fairness bound. However, if the term $\frac{2B}{\mu}$ is large, the overall improvement in group fairness may be limited, as this dominant term can overshadow the effects of reducing $\epsilon$.

On the other hand, increasing $\epsilon$ can have adverse effects on subgroup fairness since $\epsilon^2 \cdot \Gamma$ will increase, leading to greater subgroup discrepancies. In terms of group fairness, an increase in $\epsilon$ can potentially yield improvements if the other terms, such as $\frac{\kappa}{\gamma+\Gamma-1}$ and $\frac{2B}{\mu}$, dominate the fairness bound. However, this benefit is constrained if $\epsilon^2 \cdot \Gamma$ is already a small component within the bound.

Balancing these factors involves a trade-off between minimizing subgroup discrepancies and optimizing group fairness, considering the relative magnitudes of the different terms in the fairness bound. Careful tuning of $\epsilon$ is essential to achieve a desirable balance that promotes both subgroup and group fairness.

The trade-off between subgroup and group fairness can be managed by carefully tuning $\epsilon$ while considering the effects of $\kappa, \gamma, \mu$, and $B$. The goal is to find an optimal value of $\epsilon$ that minimizes both subgroup and group discrepancies within acceptable limits. This involves balancing the impact of these parameters to avoid disproportionately favoring one type of fairness over the other.

### C.3.1 DOMINANCE OF TERM $\epsilon$

When the other parameters ($\kappa$, $\gamma$, $\Gamma$, B, and $\mu$) are fixed without dominance, the primary variable affecting the fairness bounds is $\epsilon$. In this scenario, if $\epsilon^2$ dominates the other terms in the fairness bounds, then reducing $\epsilon$ will have a significant impact on both subgroup and group fairness bounds; reducing $\epsilon$ can simultaneously improve both subgroup fairness and group fairness, as the $\epsilon^2$ terms are reduced in both bounds. Thus, under the assumption that $\epsilon^2$ is the dominant term and other terms are fixed, it is possible for there to be no significant trade-off between subgroup fairness and group fairness. However, **in practical scenarios**, the other terms may still exert influence, and the interdependence between $\epsilon$ and the constants ($\kappa$, $\gamma$, $\Gamma$, and $\beta$) can lead to a trade-off. While $\epsilon^2$ may play a crucial role, the practical interactions of all parameters need consideration to fully understand fairness dynamics. In these specific conditions the trade-off between subgroup and group fairness might be minimized or even eliminated. By highlighting these scenarios, we aim to provide a more comprehensive understanding of how the dominance of $\epsilon^2$ can significantly influence fairness outcomes, thereby offering practical guidance for optimizing fairness in FL models.

### C.4 PRIVACY ANALYSIS

In this section, we present a detailed mathematical analysis of how differential privacy (DP) is applied in LipFed to protect subgroup losses and fairness constraints while maintaining model utility. The goal is to ensure that sensitive data remains private without compromising the ability to mitigate subgroup bias.

### C.4.1 DIFFERENTIAL PRIVACY IN LIPFED

Differential privacy ensures that the inclusion or exclusion of a single data point (or client) does not significantly affect the outcome of the computation, thereby protecting sensitive data. LipFed integrates DP by adding *Laplace noise* to the local subgroup losses, ensuring privacy in the exchange of fairness-related metrics between clients and the server.

**Definition of Differential Privacy.** A randomized algorithm $A$ satisfies $\epsilon$-differential privacy if, for any two adjacent datasets $D$ and $D'$ (differing by only one data point), and for any set $S$ of possible outputs:

$$P(A(D) \in S) \leq e^\epsilon \cdot P(A(D') \in S)$$

where $\epsilon$ is the *privacy budget*, controlling the amount of noise added and the trade-off between privacy and accuracy.

### C.4.2 Applying Differential Privacy to Subgroup Losses

In LipFed, we introduce Laplace noise to the local subgroup losses to maintain privacy. The randomized mechanism for applying DP to subgroup losses is defined as:

$$A(D) = \hat{R}(X_g; \theta) + \text{Laplace}\left(\frac{\Delta R}{\epsilon}\right) \tag{26}$$

where $\hat{R}(X_g; \theta)$ is the true risk or loss function for subgroup $X_g$; $\Delta R$ is the sensitivity of the loss function, measuring the maximum change in output by modifying a single client's data; $\epsilon$ is the privacy budget controlling the amount of noise added.

### C.4.3 Sensitivity of Subgroup Losses

The *sensitivity* $\Delta R$ of the loss function is the maximum possible difference in the loss function due to the change in one client's data. If $R(X_g; \theta)$ represents the loss for subgroup $X_g$, then:

$$\Delta R = \max_{D, D'} |R(D; \theta) - R(D'; \theta)| \tag{27}$$

where $D$ and $D'$ are neighboring datasets differing by only one data point.

### C.4.4 Noise Addition and Privacy Guarantee

For each subgroup, we add Laplace noise $\text{Laplace}\left(\frac{\Delta R}{\epsilon}\right)$ to ensure that the differences in the subgroup losses remain indistinguishable. The magnitude of the noise is proportional to the sensitivity $\Delta R$ and inversely proportional to $\epsilon$, where larger $\epsilon$ implies less noise and weaker privacy guarantees.

This ensures that the exchange of sensitive subgroup performance information between the server and clients is protected by differential privacy.

### C.4.5 Impact on Fairness and Utility

The introduction of DP in LipFed does not significantly degrade fairness or model utility, as seen in the experimental results. For instance, different privacy budgets $\epsilon \in \{0.8, 1.0, 1.4\}$ only minimally affect subgroup fairness and overall accuracy.

**Theoretical Privacy Bound.** LipFed ensures that the discrepancy between the loss values of similar subgroups is bounded by $\epsilon$-differential privacy. Given the Lipschitz continuity constraint $D(h_\theta(X), h_\theta(X')) \leq \epsilon \cdot d(X, X')$, we enforce that:

$$|R(X_g; \theta) - R(X_g'; \theta)| \leq \epsilon^2 \cdot \Gamma \tag{28}$$

where $\Gamma$ measures the heterogeneity in data distribution across clients. This bound ensures that subgroup discrepancies remain within the privacy budget while preserving fairness.

## D More Related Work

FL algorithms aimed at achieving a globally fair model are typically classified into three distinct categories, including *client-fairness* Li et al. (2019a); Mohri et al. (2019); Deng et al. (2020); Li et al. (2020); Hu et al. (2022); Horvath et al. (2021), *group-fairness* Yue et al. (2021); Cui et al. (2021); Papadaki et al. (2022); Selialia et al. (2023), and *robustness techniques* Lee et al. (2022); Karimireddy et al. (2020).

## D.1 CLIENT FAIRNESS IN FEDERATED LEARNING

Ensuring fairness among clients in FL is vital to counteract biases from non-IID data distributions across devices. Techniques like the Federated Fair Averaging (FedFV)Wang et al. (2021) adjust gradient directions and magnitudes to balance model *average performance* based on each client's conflict level and contributionPapadaki et al. (2022). GIFair-FL Yue et al. (2021) dynamically adjusts model updates using a fairness-aware aggregator to reduce *average loss* across clients, while FjORD Horvath et al. (2021) employs ordered dropout to tailor model sizes to clients' device capacities, enhancing fairness and accuracy.

Additional approaches that build upon these fairness-enhancing techniques include Agnostic Federated Learning (AFL)Mohri et al. (2019), which optimizes the global model against any potential target distribution by accommodating unknown distribution mixes among clients. q-FFLLi et al. (2019a) addresses data heterogeneity by reweighting losses to prioritize devices with poorer performance, promoting uniform model accuracy across devices. Tilted empirical risk minimization (TERM) Li et al. (2020) adjusts the influence of outliers and balances class representation through a flexible tilt hyperparameter. These methods enhance *average performance* in FL systems operating in heterogeneous environments.

## D.2 GROUP FAIRNESS IN FEDERATED LEARNING

Recent advancements in FL have highlighted the importance of addressing fairness concerns, particularly group fairness, where biases against protected demographic groups are mitigated. Ezzeldin et al. (2023) introduced FairFed, a strategy that ensures fair model training by employing a fairness-aware aggregation method. In FairFed, each client performs local debiasing using their own dataset to maintain decentralization and privacy. Clients evaluate the global model's fairness in each FL round, and aggregation weights are adjusted in collaboration with the server based on the mismatch between global and local fairness metrics. This method, supported by secure aggregation protocols, enhances group fairness under heterogeneous data conditions and allows for client-specific debiasing techniques, showing significant improvement over traditional fairness approaches in FL settings. FairFed's empirical validation confirms its effectiveness in achieving group fairness, with plans for future enhancements to accommodate various application scenarios and integrate broader fairness concepts, such as collaborative and client-based fairness.

In a parallel effort, Papadaki et al. (2022) explore group fairness in FL through their FedMinMax algorithm, which is crafted to establish minimax fairness across demographic groups, an approach that differs from traditional methods aimed at equalizing performance across clients. FedMinMax strategically employs alternating optimization techniques—projected gradient ascent for optimizing weights and stochastic gradient descent for the model—tailoring the learning process to balance fairness among demographic groups effectively. This method has demonstrated competitive or superior performance against established benchmarks in various FL setups, showcasing its capability to uphold group fairness robustly. Simultaneously, Cui et al. (2021) propose the FCFL framework, which addresses both algorithmic fairness and performance consistency across distributed data sources in FL. Derived from a constrained multi-objective optimization perspective, FCFL aims to maximize the utility of the least advantaged client while meeting fairness constraints, achieving Pareto optimality via gradient-based methods. Theoretical and empirical validations of FCFL underscore its ability to outperform existing models in ensuring fairness and consistent performance across clients, making it a viable solution for real-world applications where these attributes are crucial. These developments collectively signal a shift towards more ethical and equitable federated learning environments, emphasizing the necessity for continuous innovation in fairness-oriented methodologies within the field.

## D.3 ROBUSTNESS IN FEDERATED LEARNING

The paper Lee et al. (2021) addresses the challenge of data heterogeneity and forgetting in federated learning (FL), where a global model is collaboratively learned without direct access to clients' data. Drawing an

analogy to continual learning, the study proposes that forgetting could hinder FL's convergence. They observe that the global model forgets knowledge from previous rounds, and local training induces forgetting outside the local distribution. The authors hypothesize that addressing forgetting could alleviate data heterogeneity issues. To tackle this, they propose Federated Not-True Distillation (FedNTD), a novel algorithm that preserves the global perspective on locally available data only for the not-true classes. FedNTD effectively mitigates forgetting and demonstrates state-of-the-art performance in various FL setups. Through empirical analysis, the study confirms that the global model's prediction consistency suffers across communication rounds due to forgetting induced by data heterogeneity. FedNTD addresses this by selectively preserving global knowledge outside local distributions, offering a promising solution to improve FL performance without compromising data privacy or incurring additional communication costs.

Table 2: Partitioning of datasets with added Gaussian noise

| Client | Samples | Noise STD | Test Data | Samples | Noise STD | Test Data | Samples | Noise STD | Test Data |
|--------|---------|-----------|-----------|---------|-----------|-----------|---------|-----------|-----------|
| | | **MNIST** | | | **FashionMNIST** | | | **FER2013** | |
| 1 | 6,000 | 0.4 | Original + 0.4 | 6,000 | 0.4 | Original + 0.4 | 2870 | 0.0 | Original + 0.0 |
| 2 | 6,000 | 0.5 | Original + 0.5 | 6,000 | 0.5 | Original + 0.5 | 2870 | 0.09 | Original + 0.09 |
| 3 | 6,000 | 0.7 | Original + 0.7 | 6,000 | 0.7 | Original + 0.7 | 2870 | 0.18 | Original + 0.18 |
| 4 | 6,000 | 1.0 | Original + 1.0 | 6,000 | 1.0 | Original + 1.0 | 2870 | 0.27 | Original + 0.27 |
| 5 | 6,000 | 1.5 | Original + 1.5 | 6,000 | 1.5 | Original + 1.5 | 2870 | 0.36 | Original + 0.36 |
| 6 | 6,000 | 0.4 | Original + 0.4 | 6,000 | 0.4 | Original + 0.4 | 2870 | 0.0 | Original + 0.0 |
| 7 | 6,000 | 0.5 | Original + 0.5 | 6,000 | 0.5 | Original + 0.5 | 2870 | 0.09 | Original + 0.09 |
| 8 | 6,000 | 0.7 | Original + 0.7 | 6,000 | 0.7 | Original + 0.7 | 2870 | 0.18 | Original + 0.18 |
| 9 | 6,000 | 1.0 | Original + 1.0 | 6,000 | 1.0 | Original + 1.0 | 2870 | 0.27 | Original + 0.27 |
| 10 | 6,000 | 1.5 | Original + 1.5 | 6,000 | 1.5 | Original + 1.5 | 2870 | 0.36 | Original + 0.36 |
| | | **UTK** | | | **ACS Income (ASCI)** | | | **ACS Employment (ASCE)** | |
| 1 | 1920 | 0.0 | Original + 0.0 | 26621 | - | State test | 6656 | - | State test |
| 2 | 1920 | 0.1 | Original + 0.1 | 11143 | - | State test | 2768 | - | State test |
| 3 | 1920 | 0.3 | Original + 0.3 | 156532 | - | State test | 39133 | - | State test |
| 4 | 1920 | 0.5 | Original + 0.5 | 32091 | - | State test | 8023 | - | State test |
| 5 | 1920 | 0.7 | Original + 0.7 | 41653 | - | State test | 10414 | - | State test |
| 6 | 1920 | 0.0 | Original + 0.0 | 108739 | - | State test | 27185 | - | State test |
| 7 | 1920 | 0.1 | Original + 0.1 | 13069 | - | State test | 3268 | - | State test |
| 8 | 1920 | 0.3 | Original + 0.3 | 12645 | - | State test | 3162 | - | State test |
| 9 | 1920 | 0.5 | Original + 0.5 | 17604 | - | State test | 4402 | - | State test |
| 10 | 1920 | 0.7 | Original + 0.7 | 17814 | - | State test | 4454 | - | State test |

# E   EXPERIMENTAL SETUP

## E.1   DATASET DETAILS

**Choice of Datasets.** In our experiments, we evaluated the LipFed framework using four small datasets, including MNIST, Fashion-MNIST, FER2013, and UTK, and two large scale dataset, including ASCI and ASCE, with a 10 clients. These datasets were chosen to represent a diverse set of applications, thereby providing a comprehensive evaluation of the feasibility and initial effectiveness of the proposed subgroup

fairness technique. Each dataset presents unique characteristics and challenges related to bias studies. The MNIST dataset consists of handwritten digit images. This dataset is often used as a benchmark for image classification tasks and serves as a starting point for evaluating model performance on simple, grayscale images. It helps in understanding basic biases that might arise from digit shapes and writing styles. Fashion-MNIST is a dataset of grayscale images of clothing items. This dataset is used to test model performance on more complex visual patterns compared to MNIST. It introduces variability in clothing styles, textures, and shapes, which can help identify biases related to visual feature extraction and classification. The FER2013 dataset contains grayscale images of facial expressions. This dataset is crucial for studying biases related to facial recognition and emotion detection. It includes images with diverse facial expressions and varying degrees of emotion intensity, which can reveal biases in recognizing and classifying emotional states, especially across different demographic groups. The UTKFace dataset includes images of faces with annotations for age, gender, and ethnicity. This dataset is particularly valuable for studying intersectional biases involving age, gender, and ethnicity. It allows for an in-depth analysis of how different demographic attributes can impact model performance and fairness, revealing potential biases in facial recognition systems across diverse population groups. Despite the aforementioned datasets, we recognize the importance of assessing the model's scalability and robustness on larger datasets, we perform further evaluations on large real-world datasets (ACSI and ACSE) which is used in fairness studies.

**Data Partitions.** As it is customary to partition benchmark datasets across clients in FL research Hsu et al. (2019); Wang et al. (2020), we adopt this strategy and distribute samples of an individual group equally across clients according to the Dirichlet distribution Hsu et al. (2019). This distribution is demonstrated in Table 2, where the third column shows that distributing samples of an individual group equally across clients leads to clients with the equal number of samples in their local data $\mathscr{D}_k$. The uniform data partitioning strategy is motivated by the desire to demonstrate that even in FL settings with balanced groups across clients, feature noise heterogeneity still leads to subgroup bias across clients.

**Heterogeneous Feature Distributions.** We introduce feature noise across data partitions to simulate real-world scenarios where images are non-IID, deviating from the feature distribution of pristine training images Ghosh et al. (2018); Saenko et al. (2010); Song et al. (2022). The noise is added to an image by adding a random value sampled from a Gaussian distribution to each pixel of the image. Mathematically, this is represented as:

$$\tilde{I}(x,y) = I(x,y) + \epsilon \tag{29}$$

where $\epsilon \sim \mathcal{N}(0, \sigma^2)$, with $\tilde{I}(x,y)$ and $I(x,y)$ denoting noisy and original pixel values at $(x,y)$, respectively. The parameter $\sigma$ controls the amount of noise added to the image. The larger the value of $\sigma$, the more intense the noise. Specifically, Gaussian noise with $\sigma$ of $0.03$ or higher is incorporated, reflecting conditions observed in real-world deployments Lyu et al. (2020). The noise addition to each client's local training dataset $\mathscr{D}_k$ is demonstrated in Table 2, where the fourth column shows all local datasets across different clients have different feature noise distributions. The difference in feature noise across clients is motivated by the desire to understand how the nonIID-ness in subgroup data of an individual group affects the global model's bias across subgroups.

**Local Test Data.** Each client utilizes a replicated version of the original benchmark test set, aligning similar noise feature distributions between the training and test data for individual clients. For example, as depicted in Table 2, client 1 employs the original FMNIST test dataset with noise levels consistent with those of the training partition. This approach is motivated by the assumption that the local and training data for each client share similar feature distributions, which may differ from those of other clients.

## E.2 TRAINING PARAMETERS

Table 3 outlines the primary training parameters used across all models and datasets in this work. We implemented the system using PyTorch pytorch on Ubuntu 22.04 (8GB NVIDIA Quadro P2200 GPU). [1]

Table 3: Model Training Parameters.

| Algorithm | Dataset | Train time per round (minutes) | Model | Minibatch size | Momentum | Weight decay | Learning rate | # Local epochs | # Rounds | Loss function |
|---|---|---|---|---|---|---|---|---|---|---|
| FedAvg | MNIST | 2.25 | LeNet | 256 | 0.9 | 0.0001 | 0.01 | 5 | 65 | Cross entropy |
| | Fashion-MNIST | 2.23 | VGGNet | 256 | 0.9 | 0.0005 | 0.01 | 5 | 65 | Cross entropy |
| | FER2013 | 8.93 | ResNet-18 | 128 | 0.9 | 0.0005 | 0.01 | 5 | 30 | Cross entropy |
| | UTK | 4.98 | ResNet-18 | 64 | 0.9 | 0.0005 | 0.01 | 5 | 75 | Cross entropy |
| | ACSIncome | - | Logistic R. | 128 | - | - | 0.001 | 5 | 10 | Binary Cross entropy |
| | ACSEmpoyment | - | Logistic R. | 128 | - | - | 0.001 | 5 | 10 | Binary Cross entropy |
| AFL | MNIST | 2.22 | LeNet | 256 | 0.9 | 0.0001 | 0.01 | 5 | 65 | Cross entropy |
| | Fashion-MNIST | 2.25 | VGGNet | 256 | 0.9 | 0.0005 | 0.01 | 5 | 65 | Cross entropy |
| | FER2013 | 8.76 | ResNet-18 | 128 | 0.9 | 0.0005 | 0.01 | 5 | 30 | Cross entropy |
| | UTK | 4.94 | ResNet-18 | 64 | 0.9 | 0.0005 | 0.01 | 5 | 75 | Cross entropy |
| | ACSIncome | - | Logistic R. | 128 | - | - | 0.001 | 5 | 10 | Binary Cross entropy |
| | ACSEmpoyment | - | Logistic R. | 128 | - | - | 0.001 | 5 | 10 | Binary Cross entropy |
| TERM | MNIST | 2.27 | LeNet | 256 | 0.9 | 0.0001 | 0.01 | 5 | 65 | Cross entropy |
| | Fashion-MNIST | 2.28 | VGGNet | 256 | 0.9 | 0.0005 | 0.01 | 5 | 65 | Cross entropy |
| | FER2013 | 9.15 | ResNet-18 | 128 | 0.9 | 0.0005 | 0.01 | 5 | 30 | Cross entropy |
| | UTK | 4.99 | ResNet-18 | 64 | 0.9 | 0.0005 | 0.01 | 5 | 75 | Cross entropy |
| | ACSIncome | - | Logistic R. | 128 | - | - | 0.001 | 5 | 10 | Binary Cross entropy |
| | ACSEmpoyment | - | Logistic R. | 128 | - | - | 0.001 | 5 | 10 | Binary Cross entropy |
| GIFAIR-FL | MNIST | 2.05 | LeNet | 256 | 0.9 | 0.0001 | 0.01 | 5 | 65 | Cross entropy |
| | Fashion-MNIST | 1.98 | VGGNet | 256 | 0.9 | 0.0005 | 0.01 | 5 | 65 | Cross entropy |
| | FER2013 | 8.27 | ResNet-18 | 128 | 0.9 | 0.0005 | 0.01 | 5 | 30 | Cross entropy |
| | UTK | 4.51 | ResNet-18 | 64 | 0.9 | 0.0005 | 0.01 | 5 | 75 | Cross entropy |
| | ACSIncome | - | Logistic R. | 128 | - | - | 0.001 | 5 | 10 | Binary Cross entropy |
| | ACSEmpoyment | - | Logistic R. | 128 | - | - | 0.001 | 5 | 10 | Binary Cross entropy |
| FedNTD | MNIST | 2.53 | LeNet | 256 | 0.9 | 0.0001 | 0.01 | 5 | 65 | Cross entropy |
| | Fashion-MNIST | 2.57 | VGGNet | 256 | 0.9 | 0.0005 | 0.01 | 5 | 65 | Cross entropy |
| | FER2013 | 10.23 | ResNet-18 | 128 | 0.9 | 0.0005 | 0.01 | 5 | 30 | Cross entropy |
| | UTK | 5.62 | ResNet-18 | 64 | 0.9 | 0.0005 | 0.01 | 5 | 75 | Cross entropy |
| | ACSIncome | - | Logistic R. | 128 | - | - | 0.001 | 5 | 10 | Binary Cross entropy |
| | ACSEmpoyment | - | Logistic R. | 128 | - | - | 0.001 | 5 | 10 | Binary Cross entropy |
| Scaffold | MNIST | 0.71 | LeNet | 256 | 0.9 | 0.0001 | 0.01 | 5 | 65 | Cross entropy |
| | Fashion-MNIST | 0.82 | VGGNet | 256 | 0.9 | 0.0005 | 0.01 | 5 | 65 | Cross entropy |
| | FER2013 | 2.29 | ResNet-18 | 128 | 0.9 | 0.0005 | 0.01 | 5 | 30 | Cross entropy |
| | UTK | 1.63 | ResNet-18 | 64 | 0.9 | 0.0005 | 0.01 | 5 | 75 | Cross entropy |
| | ACSIncome | - | Logistic R. | 128 | - | - | 0.001 | 5 | 10 | Binary Cross entropy |
| | ACSEmpoyment | - | Logistic R. | 128 | - | - | 0.001 | 5 | 10 | Binary Cross entropy |
| LipFed | MNIST | 2.61 | LeNet | 256 | 0.9 | 0.0001 | 0.01 | 5 | 65 | Cross entropy |
| | Fashion-MNIST | 5.14 | VGGNet | 256 | 0.9 | 0.0005 | 0.01 | 5 | 65 | Cross entropy |
| | FER2013 | 9.94 | ResNet-18 | 128 | 0.9 | 0.0005 | 0.01 | 5 | 30 | Cross entropy |
| | UTK | 5.14 | ResNet-18 | 64 | 0.9 | 0.0005 | 0.01 | 5 | 75 | Cross entropy |
| | ACSIncome | - | Logistic R. | 128 | - | - | 0.001 | 5 | 10 | Binary Cross entropy |
| | ACSEmpoyment | - | Logistic R. | 128 | - | - | 0.001 | 5 | 10 | Binary Cross entropy |

---

[1]The 'readme.txt' file at the root of the project folder consists of the steps required to run the code: Download Zipped Folder

### E.3 ADAPTATION TO TABULAR DATASETS

Our approach of using the average variance of image pixels is directly applicable to tabular data. We first present a detailed methodology for adapting LipFed for two tabular datasets from fair ML Retiring Adult datasets Ding et al. (2021), ACSIncome and ACSEmployment:

The steps to compute subgroup weights/moments (e.g., variance) for subgroups are as follows:

1. Data Separation: Divide data into subgroups based on intersecting attributes (e.g., income >50K and demographic areas).
2. Variance Calculation: Calculate variance (subgroup weight) for each subgroup: $\sigma_g^2 = \frac{1}{N_g} \sum_{i=1}^{N_g} (x_i - \mu_g)^2$.

Here, $N_g$ is the number of samples in subgroup $g$, $x_i$ are the feature values, $\mu_g$ is the mean of the feature for subgroup $g$, and $\sigma_g$ is the standard deviation of the feature for subgroup $g$. Results in Figure 1 show our approach's effectiveness in bias mitigation, even for tabular data.

We use the ACS PUMS Ding et al. (2021) as the basis for both prediction tasks income and employment:

**Example: ACSIncome Prediction.** We use ACS PUMS data to gather income-related features, race, and state information, ensuring each data point includes the state it belongs to. Data is distributed across clients based on the state attribute (randomly selected USA states), with each client representing data from a specific state. We define two income groups:

1. Income True: Individuals with income above a certain threshold (e.g., $50,000).
2. Income False: Individuals with income below this threshold.

The state serves as an *implicit sensitive attribute* due to its correlation with demographic distribution, forming subgroups by income level and demographic region (e.g., Income True and California).

**Example: ACSEmployment Prediction.** For ACSEmployment, the task is to predict whether an individual is employed after filtering ACS PUMS data to include individuals between the ages of 16 and 90 Ding et al. (2021);. We define two employment groups:

1. Employed: Individuals who are currently employed.
2. Unemployed: Individuals who are not employed.

The steps to compute subgroup weights for this dataset are similar: Divide data into subgroups based on employment status and demographic attributes (e.g., employed and from California). Compute variance for each subgroup as described earlier, allowing us to weigh the subgroups' importance and enforce subgroup fairness in the optimization problem discussed in Section 4.1 of the paper.

This methodology illustrates the adaptability of the LipFed framework to diverse data types, emphasizing its utility in addressing fairness across multiple domains.

## F METRICS

### F.1 TRUE POSITIVE RATE (TPR)

The True Positive Rate (TPR) is a critical metric for assessing model performance, as it measures the proportion of actual positives correctly predicted by the model. Variations in TPR across subgroups indicate

discrepancies in the model's generalization across different subpopulations. In FL, TPR variation is often a result of non-IID data across clients. Subgroups with diverse characteristics—such as demographic differences, sensor quality, or geographical factors—lead to varied feature distributions, causing differential model performance. Mathematically, TPR is defined as:

$$TPR_g = \frac{TP_g}{TP_g + FN_g} \tag{30}$$

where $TP_g$ and $FN_g$ represent the true positives and false negatives for subgroup $g$, respectively.

### F.1.1 Variation in TPR Across Subgroups

The variation in TPR can quantify the performance discrepancies between subgroups. Let the TPR for each subgroup $g$ be denoted as $TPR_g$. The difference between the highest can measure the discrepancy in performance among subgroups- and lowest-performing subgroups:

$$Disc(TPR) = \max_g(TPR_g) - \min_g(TPR_g) \tag{31}$$

A large discrepancy suggests that some subgroups benefit more from the model than others, highlighting the presence of subgroup bias. In non-IID FL settings, subgroup $g$ on one client may have very different feature distributions compared to the same subgroup on another client, leading to inconsistent TPRs across subgroups.

In our LipFed framework, which applies Lipschitz constraints to reduce subgroup bias, the goal is to minimize the performance discrepancy across subgroups. The performance difference is constrained by a Lipschitz continuity condition that controls how much the TPR can vary based on subgroup similarity. This condition ensures that:

$$D(h_\theta(x), h_\theta(x')) \leq \epsilon \cdot d(x, x') \tag{32}$$

where $D(h_\theta(x), h_\theta(x'))$ represents the Euclidean distance between the model's outputs for two subgroup instances $x$ and $x'$, and $d(x, x')$ is a distance metric quantifying the similarity between the subgroups.

Thus, variations in TPR among subgroups are restricted by the parameter $\epsilon$, which limits the magnitude of subgroup performance differences:

$$Disc(TPR) \leq \epsilon \tag{33}$$

By enforcing these Lipschitz constraints, LipFed reduces the subgroup performance disparity, resulting in more equitable TPRs across clients.

### F.2 Median/Average Performance Discrepancy

The maximum discrepancy metric focuses on the largest performance gap between subgroups, which can highlight the worst-case unfairness. We acknowledge that relying solely on the maximum performance discrepancy among all subgroups may not always provide a complete picture of model fairness; this approach may unfairly penalize a model that performs exceptionally well for most subgroups but poorly for one specific subgroup. To provide a more comprehensive evaluation of model fairness, we used the median/average performance discrepancy across all subgroups to provide a more balanced view of fairness as reported in §5. Median/average discrepancy provides a more balanced view of the model's performance across all subgroups, commonly used in FL group fairness studies such as Poulain et al. (2023); Yue et al. (2023). It accounts for the median/average difference between subgroup performances rather than just focusing on the worst-case scenario. By considering the median/average performance difference, the sensitivity to outliers that might disproportionately affect the maximum discrepancy metric is reduced. In summary, the maximum performance discrepancy among all subgroups may not always provide a complete picture of model fairness.

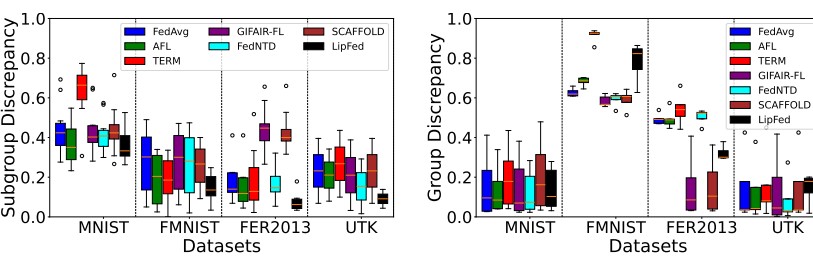

(a) TPR variations among subgroups.     (b) TPR variations among groups.

Figure 8: Demonstrating subgroup bias in model performance for different datasets and baselines.

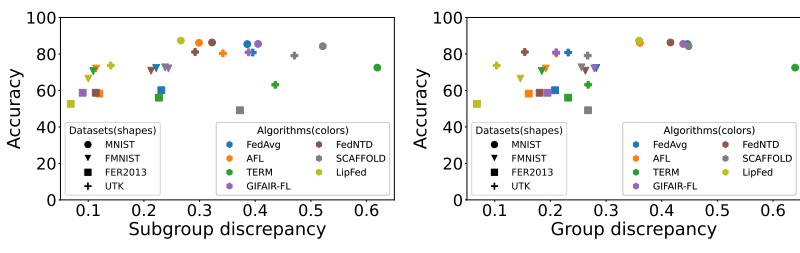

(a) Utility vs. Subgroup fairness.     (b) Utility vs. Group fairness.

Figure 9: Demonstrating model utility vs. discrepancy for different datasets and baselines.

## G    ADDITIONAL RESULTS

### G.1    LIPFED'S PERFORMANCE AGAINST CONSISTENCY AND ROBUSTNESS BENCHMARKS

In addition to fairness benchmarks (AFL, TERM, GIFAIR), we compare LipFed, our fairness-focused technique, against FL algorithms such as Scaffold Karimireddy et al. (2020) and FedNDT Lee et al. (2022), which prioritize robustness and consistency over fairness. In the *FL heterogeneity category*, FedNTD addresses performance loss due to data heterogeneity by managing global model memory. In the *FL robustness category*, SCAFFOLD Karimireddy et al. (2020) focuses on enhancing resilience against outliers and noisy data, mitigating the impact of irregularities in local datasets. Scaffold addresses client drift, while FedNDT targets model discrepancies due to non-IID data. This evaluation measures LipFed's performance in reducing subgroup bias and maintaining model utility compared to these non-fairness benchmarks.

The results indicate that while Scaffold and FedNDT exhibit strong robustness and consistency across various datasets, LipFed outperforms both in terms of reducing subgroup bias, as shown in Figure 8. For example, in the MNIST dataset, LipFed achieves a 20% lower subgroup bias than Scaffold, demonstrating its effectiveness in mitigating bias without sacrificing much performance. Importantly, although LipFed is designed to mitigate subgroup bias, it also improves group fairness, showing reductions in group discrepancy similar to those seen in subgroup fairness. This demonstrates that LipFed's benefits extend beyond subgroup bias mitigation.

Additionally, in Figure 9, we show that LipFed maintains competitive performance across all datasets, with trends in utility closely mirroring those of the robustness-focused methods. While Scaffold and FedNDT slightly outperform LipFed in raw performance metrics, the trade-off is minimal, showcasing that LipFed effectively balances both fairness and performance across diverse data conditions.

This comparison highlights that while methods like Scaffold and FedNDT excel in providing robustness, LipFed offers a balanced solution by significantly reducing subgroup bias while still maintaining strong performance across diverse datasets.

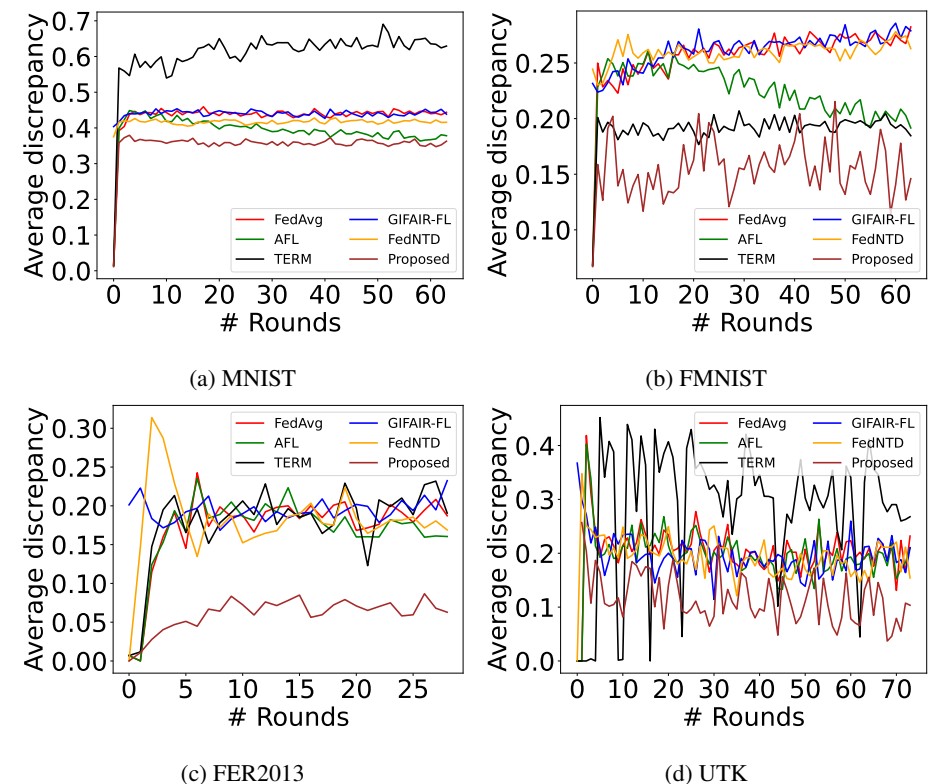

(a) MNIST

(b) FMNIST

(c) FER2013

(d) UTK

Figure 10: Convergence of the training subgroup discrepancy of LipFed and other baseline techniques across multiple datasets. LipFed consistently exhibits lower subgroup discrepancy across all iterations.

## G.2 CONVERGENCE ANALYSIS

We evaluate the convergence behavior of LipFed in comparison to baseline techniques such as AFL, TERM, and GIFAIR. The goal is to assess how quickly the training process reduces subgroup discrepancies across multiple datasets, including MNIST, Fashion-MNIST, FER2013, and UTK. Convergence here refers to the stability and speed at which the subgroup discrepancy is minimized during the training process.

As shown in Figure 10, LipFed consistently demonstrates faster convergence and lower subgroup discrepancy across all datasets. This rapid reduction in subgroup bias is primarily due to the Lipschitz continuity constraints imposed by LipFed, which ensure that performance differences between subgroups are bounded early in the training process. In contrast, the baseline techniques either converge more slowly or stabilize at higher subgroup discrepancy values, highlighting their inability to efficiently address subgroup fairness in non-IID settings. For example, on the FER2013 dataset, LipFed achieves a 30% reduction in subgroup discrepancy within the first 50 iterations compared to AFL, which converges much slower. Similarly, on the UTK dataset, LipFed stabilizes subgroup fairness more effectively than other methods, reaching a lower discrepancy in fewer iterations. This consistent performance across datasets illustrates LipFed's efficiency in addressing fairness concerns in federated learning environments with heterogeneous client data. In summary, LipFed's convergence behavior demonstrates its ability to quickly and efficiently reduce subgroup discrepancies, outperforming other fairness-focused techniques in both speed and effectiveness.

