# OpenReview forum: "LipFed: Mitigating Subgroup Bias in Federated Learning with Lipschitz Constraints"
_ICLR.cc/2025/Conference — ICLR 2025 Conference Withdrawn Submission_

### Official Review · Reviewer_s158 · 2024-10-30

**Soundness:** 2
**Presentation:** 2
**Contribution:** 3
**Rating:** 3
**Confidence:** 4

**Summary:**

The paper addresses the problem of subgroup fairness in federated learning (FL). The motivation is that previous algorithms designed for group fairness in FL may be unable to achieve subgroup fairness across clients, due to a potential feature skew across clients which makes subgroups non-IID in terms of their feature distribution. Hence, ensuring fairness across both intersectional subgroups and broad groups is necessary. To achieve this goal, the work ensures equitable model performance across diverse subgroups by adding a regularization term to the objective function of FedAvg, which penalizes high discrepancy in clients' subgroup losses. By experimental results, they show that the algorithm can reduce subgroup unfairness in FL settings.

**Strengths:**

There are some strengths in the paper, listed below:
* The considered problem is an interesting problem, which seems to be not studied before.
* An extensive discussion of the existing related works on group/subgroup fairness in centralized/FL settings is done in the appendix.
* The proposed idea is evaluated on four datasets and multiple baseline algorithms.

**Weaknesses:**

Despite the mentioned strengths, the work has multiple weaknesses, as listed below. I will clarify further about them in my questions.
* The writing of the paper needs to be improved, as there are multiple typos, ambiguities and few wrong claims.
* The proposed algorithm, LipFed, is addressing an interesting problem, but it has a heuristic nature:  it introduces multiple parameters, more precisely $\epsilon, t$ (the regularization weight), $w_{g,k}$ (the importance weights) with limited heuristic methods for setting them, which makes me hesitate about the applicability of the proposed approach in real scenarios. The authors have mentioned this in the appendix as a limitation of the work.
* LipFed, induces multiple constraints that may not be addressable for different models and datasets. This also induces computational overheads, which the authors have mentioned in the appendix as a limitation. A study of the induced computational overhead compared to that of simpler algorithms, e.g., FedAvg, will make it more clear whether the overhead is tolerable or not.

* The theoretical analysis in Theorem 4.2.2 barely delivers a clear message.
* The experimental evaluations need to be improved to include some other baselines and some ablation studies.

I will ask more detailed questions in the following to clarify about my opinion.

**Questions:**

In the following, I highlight the questions that are more important to me. A short answer, in one or two lines, will suffice for the other questions.

1. Although the authors start LipFed's proposal with discussion about Lipschitz property, in practice it is not used in LipFed at all. What LipFed does in practice is regularizing the objective function of FedAvg with a measure of loss discrepancy across subgroups (for every existing group). As stated in the paper, the main barrier for using the Lipschitz property between two subgroups $g_k$ and $g_{k'}$ (of a group $g$) is that there is no way in FL to measure dissimilarity of features between the two subgroups.

2. (important) In equation (6), where the subgroup fairness constraint is introduced,  the goal is to make the subgroup losses of the existing $K$ clients close to each other (for all existing groups $g$). This is done by penalizing the discrepancy of clients' subgroup losses from the average subgroup loss (on the same group). The importance weights $w_{g,k}$ assigned to client $k$ is equal to the inverse of the average feature variance in the subgroup (client) $k$ (the feature corresponding to the group $g$), as stated in line 288. However, as you have clearly mentioned in Example 1, the intra-client feature variance can be irrelevant to inter-client feature variance: in example 1, and for the group $g=$"women", the variance in each subgroup $g_1$ (client 1) and $g_2$ (client 2) is small, while the variance between the two subgroups is large (from mostly white images in client 1 to mostly black images in client 2). Does LipFed assign a larger importance weight to clients 1 and 2 compared to another third client which has a mix of white and black images (i.e. has a higher intra-client feature variance)? This method of setting $w_{g,k}$ seems heuristic to me. What if the data distribution in each client is iid, but it is non-iid across clients with high variance? This challenge for setting the weights rises from the limitation that we can not measure feature dissimilarity across clients in FL.


3. (important) Other than the importance weights above, $\epsilon$ and $t$ need to be set "carefully", and the algorithm is sensitive to them. An ablation study on these two parameters to show the sensitivity of LipFed's performance to them will be clarifying.

4. (important) In the experimental results, only the GiFair baseline is on group fairness. TERM and AFL baselines are on client fairness. FedAvg is neither group-fair nor client-fair. LipFed is focusing on subgroup fairness, and needs to be compared to both the above categories to study whether it induces any costs on client fairness or group fairness. Comparison to more complex algorithms, e.g., FCFL [1], is vital. FCFL addresses both client and group fairness by min-max optimization across clients and enforcing "Equal Opportunity" (equal TPR) across sensitive attributes (groups). In contrast, LipFed does minimization across clients to minimize the average client loss regularized by subgroup loss discrepancy, eq (7). Hence, FCFL is the best baseline to compare with. This suggests a way to improve the experimental results of the paper.

[1] S. Cui, et.al., "Addressing algorithmic disparity and performance inconsistency in FL", 2021.


5. (important) Another important result that can provide a clear insight about LipFed is an ablation study on the feature skew (non-IIDness) across subgroups. This can be done by an ablation study on the noise variance $\sigma^2$ (in line 186 and Table 2 in the appendix). This is also related to question 2, above.


6. The statement of Theorems 4.2.1 and 4.2.2 need to be improved and more precise. For example, in Th. 4.2.1, where is the model $h$ coming from? from LipFed? Also, theorem 4.2.2 barely delivers a clear message to me. LipFed is not addressing group fairness at all in its objective function, and just focuses on subgroup fairness. How come can it provide any guaranty on the group fairness of the final model?

7. What is $G$ (the number of groups) in your experiments? More precisely, how many sensitive attributes, e.g., race, gender, exist in your datasets?

Minor comments to improve the writing of the paper:
* line 122: it should be $Y_k$ (not $Y_k^n$)
* line 123: the number of samples in group $g$ at client $k$ should depend on $k$. I think $N_g$ should be changed to something like $N_{g,k}$.
* line 134 (or 135): it is $R_k$ (not $F_k$).
* eq (1): $\theta_k$ should change to $\theta$ (one model parameter is learned and all clients use the same model parameter)
* line 141: the number of subgroups in a group $g$ should depend on $g$. So $n_k$ should be changed to something like $n_g$.
* line 144: both $a_1^{g_{g,k}}$ and $a_2^{g_{g,k}}$ need to be changed to something like $a_1^{g,k}$ and $a_2^{g,k}$ to be compatible with eq (2). Also, $k$ changes from 1 to $n_g$ (defined above).
* In many lines, the latex command \cite{} needs to be changed to \citep{}. For example, in line 149 (or 150).
* subgroup fairness and group fairness metrics need to be edited in lines 195 and 199,  following g the suggested notation modifications above.
* line 215, there is a typo.
* line 239, there is a typo.
* eq (6): on the right side of the inequality, $g$ should change from 1 to $G$ (not $n_g$). Also, before the inequality, $k^{i}$ should change to $k'$.
* line 281: $w_{g,k}$ should change to $w_{g,k'}$
* in Fig 6, Y axis label should change to "subgroup discrepancy".

All in all, the considered problem is interesting, but the draft needs to be improved.

---

### Official Review · Reviewer_voY5 · 2024-11-03

**Soundness:** 3
**Presentation:** 2
**Contribution:** 3
**Rating:** 5
**Confidence:** 3

**Summary:**

This paper presents a novel approach, LipFed, to address subgroup bias in federated learning by applying Lipschitz constraints. The problem addressed in this work is interesting and novel, tackling an important fairness issue in FL systems. Theoretical analysis provides solid foundations for the proposed method, and empirical results demonstrate significant improvements in subgroup fairness without major utility losses. However, incorporating comparisons with more recent baselines, expanding ablation studies, and refining the structure and detail in certain sections would further strengthen the paper’s impact and clarity.

**Strengths:**

1.	The paper addresses a novel problem by applying Lipschitz constraints to mitigate subgroup bias in federated learning.
2.	The theoretical analysis in this paper is solid, with well-defined bounds and comprehensive proofs that enhance the credibility of the proposed approach to achieving fairness in federated learning.

**Weaknesses:**

1.	The paper utilizes baseline methods AFL (Mohri et al., 2019), TERM (Li et al., 2020), and
GIFAIR-FL (Yue et al., 2021). However, recent years have seen advancements in federated
learning fairness and efficiency. It would strengthen the paper to include comparisons with
more recent baseline methods.
2.	Paper structure: Some parts of the paper are redundant, but some important parts are not clearly explained, and the method section of the article is too thin.

**Questions:**

The theoretical guarantees support LipFed’s fairness, but additional results under different
client settings would strengthen the paper. Additionally, I’m curious to see the results with
more clients, if possible.

---

### Official Review · Reviewer_NSYu · 2024-11-04

**Soundness:** 2
**Presentation:** 2
**Contribution:** 2
**Rating:** 3
**Confidence:** 4

**Summary:**

This paper addresses the challenge of mitigating *subgroup* bias in federated learning (FL). The authors begin by introducing the concept of *subgroup* fairness in the FL setting, distinguishing it from the more widely studied topic of *group* fairness in FL. They then propose LipFed, a technique to mitigate subgroup bias by incorporating Lipschitz-based fairness constraints into the learning process. The paper also provides upper bounds on group and subgroup fairness within the LipFed optimization framework. Experimental results across multiple datasets demonstrate that LipFed effectively reduces subgroup bias with minimal impact on overall utility, while also revealing a trade-off between subgroup and group fairness.

**Strengths:**

1) While most bias mitigation methods in FL focus on addressing group bias, LipFed is a novel technique designed specifically to mitigate subgroup bias in FL.

2) LipFed introduces the integration of Lipschitz-based constraints into the learning process in an FL setting, creating a compelling link between individual fairness (Dwork et al., 2012) and subgroup fairness in FL.

3) LipFed is versatile and can be readily applied to other FL algorithms, such as TERM and AFL, to reduce subgroup bias. Empirical results demonstrate its effectiveness in subgroup bias reduction.

**Weaknesses:**

1) While LipFed is designed to address subgroup bias in FL, the main text does not clearly explain how it preserves client privacy. Although Section 5.5 shows that LipFed achieves similar subgroup discrepancy and average accuracy as non-DP LipFed when tested with varying $\epsilon$ values, this result is based on experiments with only two datasets. There is no theoretical guarantee that LipFed will preserve privacy in the general case (in terms of differential privacy). Specifically, it is unclear how client $k$ accesses $R(X_{k'}^g; \theta)$. How does client $k$ compute this loss without access to the subgroup data of other clients?

2) Theoretical results, particularly Theorem 4.2.2, are informative but lack clarity regarding their implications and the relationship between subgroup and group bias. While the theorem provides an upper bound for group bias under specific assumptions, it is unclear how this result adds value to the paper or aids in understanding. A section explaining the precise implications of this theorem would be beneficial.

3) The main text contains several inconsistent and incorrect notations, some of which I highlighted in the questions section. These errors make it difficult to follow the paper’s main contributions and should be addressed for clarity.

**Questions:**

Here are some of my questions and comments. I would be willing to increase my score if my concerns are addressed:

1) Some notation inconsistencies are confusing. Unifying the notation would improve readability and make the paper easier to follow:

Line 122: $y_k \in Y_k^n$ should be $y_k^n \in Y_k$.
Line 134: $F_k$ should be $R_k$.
Lines 141–145: The distinction between $n_k$ and $N_k$ is unclear, and they seem to be used interchangeably.
Lines 145 and 196: The indexes in $Disc({a^{g,k}})$ are sometimes over $g$ and sometimes over $k$. Please clarify in the text what these indexes represent.
Line 237: Are $X_k^g$ and $X_{k'}^g$ referring to individuals or subgroups? Definition 3.1 seems to apply to individuals, so this distinction would be helpful.

2) What does $I(x,y)$ in line 186 represent? Does this imply that noise is added to all elements of the input, including the features and labels?

3) Why is the median used to compute the measure of unfairness? In fairness literature, fairness metrics are typically defined based on the worst-case scenario (e.g., demographic parity, equalized odds).

4) In Figure 7, LipFed provides both the best group and subgroup fairness for some datasets (e.g., ACSE) but achieves the best subgroup fairness with the worst group fairness for others (e.g., ACSI). Do you have any intuition about why these differences occur across datasets?

---

### Official Review · Reviewer_dsd6 · 2024-11-06

**Soundness:** 2
**Presentation:** 1
**Contribution:** 3
**Rating:** 3
**Confidence:** 4

**Summary:**

In this work the authors focus on the problem of subgroup fairness in the context of federated learning. The authors provide a new algorithm relying Lipschitz-based fairness constraints. The method comes with provable fairness guarantee and empirical performance seems good on real world datasets.

**Strengths:**

- I think the study of subgroup fairness is interesting in the context of federated learning.
- The proposed method, which borrows idea from individual fairness has some potential.

**Weaknesses:**

- Notations and equations are confusing.
1. E.g. $R_k$ in Equation (1) is defined without explanation. I assume the authors mean $F_k$.
2. Definition 3.1, $\Delta(A)$ is not defined anywhere.
3. Equation 4 is also confusing. The objective to be optimized only contains $X_k$. Shouldn't it contain all $k$ and the constraint contains all $k,k'$ pairs?
4. Equation 5, missing brackets. Also shouldn't it be $k'$ instead of $k$ inside the function $D$?
5. Equation 6, somewhere it's $k^i$, somewhere it's $k'$.
- Equation 6 seems different from authors' claim in L279-280 ("the difference between the loss of a subgroup on client k and the aggregated losses of the same subgroup across other clients k′ is small"). Equation 6 is enforcing a universal bound $\epsilon$ for the *sum over all groups* of these differences. Could the authors explain that?
- Theorem 4.2.1 could be vacuous. $\epsilon$ is a parameter that controls the loss, $\Gamma$ characterizes heterogeneity via loss. Hence the upper bound is a product of two parameter in the loss space. However, by definition, LHS is the difference in TPR, which has a naive upper bound of 1. Since there isn't any control over $R_k$, the upper bound is not even guaranteed to be smaller than 1.
- The motivation of the work is unclear. From my understanding, subgroup information is unknown therefore you can't directly apply prior group fair FL algorithm to the subgroups directly? Further, are the authors aiming to protect local subgroup fairness (achieving fair prediction on each subgroup at each local client), or global subgroup fairness (achieving fair prediction on each subgroup across the entire network)? In both case, since subgroup information is not known ahead, how do you evaluate that the subgroup fairness is achieved?
- Missing comparisons with a lot of group fair FL baselines in the experiment section, including FCFL [1], FedFair [2], FairFed [3], FedFB [4], PFFL [5], etc.
- The authors only seems to measure on Equal Opportunity. How about other group fairness metrics such as Demographic Parity and Equalized Odds?

[1] Cui, S., Pan, W., Liang, J., Zhang, C., & Wang, F. (2021). Addressing algorithmic disparity and performance inconsistency in federated learning. Advances in Neural Information Processing Systems, 34, 26091-26102.

[2] Chu, L., Wang, L., Dong, Y., Pei, J., Zhou, Z., & Zhang, Y. (2021). Fedfair: Training fair models in cross-silo federated learning. arXiv preprint arXiv:2109.05662.

[3] Cui, S., Pan, W., Liang, J., Zhang, C., & Wang, F. (2021). Addressing algorithmic disparity and performance inconsistency in federated learning. Advances in Neural Information Processing Systems, 34, 26091-26102.

[4] Zeng, Y., Chen, H., & Lee, K. (2021). Improving fairness via federated learning. arXiv preprint arXiv:2110.15545.

[5] Hu, S., Wu, Z. S., & Smith, V. (2024, April). Fair federated learning via bounded group loss. In 2024 IEEE Conference on Secure and Trustworthy Machine Learning (SaTML) (pp. 140-160). IEEE.

**Questions:**

See weaknesses.

---

### Note · Authors · 2024-11-19

I have read and agree with the venue's withdrawal policy on behalf of myself and my co-authors.